 elife.elifesciences.org

# Cortical regulation of cell size by a sizer cdr2p

Kally Z Pan[1][†], Timothy E Saunders[2,3,4][†], Ignacio Flor-Parra[1][†], Martin Howard[5]*, Fred Chang[1]*

[1]Department of Microbiology and Immunology, Columbia University College of Physicians and Surgeons, New York, United States; [2]Cell Biology and Biophysics Unit, European Molecular Biology Laboratories, Heidelberg, Germany; [3]Mechanobiology Institute, National University of Singapore, Singapore, Singapore; [4]Department of Biological Sciences, National University of Singapore, Singapore, Singapore; [5]Computational and Systems Biology, John Innes Centre, Norwich, United Kingdom

**Abstract** Cells can, in principle, control their size by growing to a specified size before commencing cell division. How any cell actually senses its own size remains poorly understood. The fission yeast *Schizosaccharomyces pombe* are rod-shaped cells that grow to ~14 µm in length before entering mitosis. In this study, we provide evidence that these cells sense their surface area as part of this size control mechanism. We show that cells enter mitosis at a certain surface area, as opposed to a certain volume or length. A peripheral membrane protein kinase cdr2p has properties of a dose-dependent 'sizer' that controls mitotic entry. As cells grow, the local cdr2p concentration in nodes at the medial cortex accumulates as a measure of cell surface area. Our findings, which challenge a previously proposed pom1p gradient model, lead to a new model in which cells sense their size by using cdr2p to probe the surface area over the whole cell and relay this information to the medial cortex.

*For correspondence: martin. howard@jic.ac.uk (MH); fc99@ columbia.edu (FC)

[†]These authors contributed equally to this work

Competing interests: The authors declare that no competing interests exist.

## Introduction

The fundamental process by which a cell controls its own size is not understood for any cell type. In actively dividing cells, growth, and size need to be coordinated for cells to maintain their size. In several cell types, cells have been shown to have a size threshold, in which they need to grow to a minimal cell size before committing to cell division (*Turner et al., 2012*). This mechanism however requires that cells somehow monitor their own size. The molecular mechanism for how size is sensed, and what aspect of size—surface area, volume, mass, linear dimensions etc—is monitored remains unknown.

The fission yeast *Schizosaccharomyces pombe* is an attractive eukaryotic model for cell size studies because of its highly regular dimensions, simple rod-shape, and growth patterns. During interphase, these cells grow from the cell tips at a nearly constant rate to approximately 14 µm in length before entering mitosis, when cell growth ceases until the next cell cycle (*Mitchison and Nurse, 1985*). Genetic analyses in fission yeast have identified a pathway of conserved protein kinases for cell size control: the DYRK kinase pom1p is an inhibitor of the SAD family kinase cdr2p, which inhibits wee1p, which in turn inhibits the cell division kinase cdk1p (*Russell and Nurse, 1987*; *Breeding et al., 1998*; *Martin and Berthelot-Grosjean, 2009*; *Moseley et al., 2009*). Loss of function of *pom1* and *wee1* leads to abnormally short cells, whereas loss of function of *cdr2* leads to abnormally long ones. Interestingly, these factors largely localize to different sites in the cell. Pom1p localizes in cortical gradients emanating from cell tips (*Bahler and Pringle, 1998*; *Padte et al., 2006*; *Hachet et al., 2011*; *Saunders et al., 2012*). Cdr2p localizes to a medial band of plasma membrane protein complexes termed 'nodes', which overlie the medial nucleus (*Morrell et al., 2004*; *Martin and*

**eLife digest** Although different types of cells come in a variety of shapes and sizes, most cells are able to maintain a fairly consistent size and shape as they grow and divide. For example, the rod-shaped cells of the fission yeast *S. pombe* grow to be 14 microns long before dividing in the middle to form two new cells. This prevents any single cell becoming too large or small.

A similar phenomenon has been observed in other types of cells, so it is clear that cells must be able to measure their own size, and then use that information to trigger cell division. A number of proteins that regulate cell size and cell division in fission yeast have now been identified. These proteins form a pathway in which a protein called pom1p inhibits another protein, cdr2p, which in turn causes a third protein, cdk1p, to start the process of cell division. However, the details of the measurement process and the property that the cells are actually measuring—surface area, volume, mass or something else—remain mysterious.

Pan et al. have now used imaging techniques and mathematical modeling to probe the distribution and movements of proteins in fission yeast cells. Their results do not support a previous model in which the cell uses the gradient of pom1p as a ruler to measure cell length. Rather, Pan et al. propose a new model in which the level of cdr2p is used to sense the size of the cell. Individual molecules of cdr2p come together to from clusters called nodes on the cell membrane. As the cell grows larger, more and more cdr2p proteins accumulate in these nodes, which are found in a band around the middle of the cell. When the cells reaches a critical cell size, the increased concentration of cdr2p at these nodes may help to trigger the start of cell division.

By examining cells that grow at different rates, Pan et al. show that the rate of accumulation of cdr2p in the nodes depends on how big the cells are, rather than on the length of time that has elapsed. Analysis of fission yeast cells of different shapes shows that cell division starts when the surface area of the cell grows to a certain value, as opposed to starting when the volume or length reach a given value.

Pan et al. also show that cdr2p binds to all parts of the cell membrane, not just to the nodes near the middle, and go on to provide a simple mathematical model showing how this property can allow cells to measure their surface area. However, as Pan et al. point out, this is probably just one component of a larger mechanism that tells cells when they need to divide.

*Berthelot-Grosjean, 2009*; *Moseley et al., 2009*). Wee1p, Cdk1p, and other regulators of mitotic entry localize primarily to the spindle pole body and nucleus (*Alfa et al., 1990*; *Masuda et al., 2011*; *Grallert et al., 2013*).

How this pathway may be used to sense cell size remains unclear. A current model for cell size control is based on pom1p concentration gradients as 'rulers' to sense cell length (*Martin and Berthelot-Grosjean, 2009*; *Moseley et al., 2009*; *Vilela et al., 2010*; *Tostevin, 2011*). This gradient model postulates that as these cells grow in length from their tips, pom1p gradients are moved away from cdr2p nodes at mid-cell, causing decreased pom1p levels at the medial cortex. This putative decrease would then allow for activation of cdr2p, leading to cdk1p activation and entry into mitosis when cells reach a critical length.

Here, we use quantitative imaging and modeling to examine the relationships of pom1p and cdr2p with cell size. We find that core assumptions of the previous pom1p gradient model are not consistent with experimental findings. We further develop a novel model in which cells monitor their size using cdr2p itself as a cortical sizer molecule to probe the surface area of the cell.

## Results

### Testing the pom1p gradient model for size control

To test the pom1p gradient model, we quantitatively analyzed pom1p in living cells expressing a functional pom1-tomato-dimer fusion protein at near-endogenous levels (*Padte et al., 2006*; *Hachet et al., 2011*; *Saunders et al., 2012*). Pom1p cortical gradients exhibit large cell-to-cell variability in intensity and distribution, fluctuate over time in individual cells, and show little consistent change with cell length (*Saunders et al., 2012*). This variability, plus a short decay length relative to cell length,

led us to question whether these gradients can function reliably as 'rulers'. One of the key predictions of the gradient-based model is that pom1p levels decrease on the medial cortex as cells grow. We measured pom1p concentration in a 3-μm region along the medial cortex, where cdr2p nodes are located. Using time-averaged data (reducing fluctuations in the gradient over time [*Saunders et al., 2012*]), we detected low but measurable intensities (*Figure 1A*, *Figure 1—figure supplement 1*). Importantly, measurements of pom1-tomato at the medial cortex showed no detectable decrease with cell length in a population of cells (*Figure 1B*), or in individual cells imaged over time (*Figure 1C,D*). These cortical measurements improve on previously reported pom1p measurements that integrate intensities over the whole cell (*Martin and Berthelot-Grosjean, 2009*; *Moseley et al., 2009*), which have artifacts stemming from the normal exclusion of pom1p from the nucleus (*Saunders et al., 2012*; *Figure 1A*, *Figure 1—figure supplement 2,3*).

To further test the gradient model, we examined the effect of altering the gradient profile. Pom1-3GFP (pom1p fused to three tandem GFPs) produced a consistently steeper gradient profile than pom1-GFP or pom1-tomato fusions (*Figure 1E,F*, *Figure 1—figure supplement 4*). The reason for this change was not clear, as these fusion proteins displayed similar dynamics (our unpublished observations). The gradient model predicts that a change in gradient distribution would lead to a significant change in cell size at division. However, we detected no differences in cell length at division between these pom1-tagged strains (*Figure 1G,H*). Consistent with this result, there were no significant differences in the intensities or number of cdr2p nodes (*Figure 1—figure supplements 4C, 5*). Overall, these data are inconsistent with the gradient model.

## Cdr2p at cortical nodes scales with cell size

To further investigate how this regulatory pathway may sense cell size, we focused on how cell size affects cdr2p and its behavior at these medial cortical nodes. Pom1p may exert its cell size effects in part by ensuring the proper localization of cdr2p nodes to this region (*Celton-Morizur et al., 2006*; *Padte et al., 2006*; *Martin and Berthelot-Grosjean, 2009*; *Moseley et al., 2009*) (see below). We quantitated cdr2p levels using a functional cdr2-GFP construct (*Figure 2A*, *Figure 1—figure supplement 1A*, *Figure 2—figure supplement 1*; *Morrell et al., 2004*). Cdr2-GFP concentration in the whole cell remained approximately constant in interphase cells of various lengths (*Figure 2B*). Interestingly, the intensity of cdr2-GFP at the medial cortex increased with cell length (*Figure 2C*, *Figure 2—figure supplements 2, 3*). The cortical area containing the nodes also increased slightly with cell length, but the relative change was less than for the cdr2-GFP intensity (*Figure 2C*; *Morrell et al., 2004*). Measurement of cdr2-GFP intensity within a 3-μm wide region of the medial cortex showed directly that the local cdr2p concentration in this region rises approximately twofold as cells grow through interphase (*Figure 2D*, *Figure 2—figure supplement 2*). This increase was confirmed in time-lapse analyses of individual cells (*Figure 2E,F*).

Time-lapse imaging also revealed dynamics of cdr2p nodes. Mature cdr2p nodes, estimated to each contain an average of ~90 cdr2-GFP molecules (*Figure 2—figure supplement 1*), moved very slowly and exhibited little change over hours (*Video 1*). FRAP studies, however, revealed that cdr2-GFP turned over with a $t_{1/2}$ of about 3 min within each node (*Figure 2—figure supplement 4*). With increasing cell length, the number of nodes in each cell increased (*Figure 2G*, *Figure 2—figure supplement 5A*), whereas the intensities of individual nodes remained unchanged (*Figure 2H*, *Figure 2—figure supplement 5B*). Thus, cell growth is accompanied by the formation of new nodes, leading to an increase in local cdr2p density. Imaging also revealed a subpopulation of less intense and more motile cortical nodes that may be newly assembling ones (*Video 1*).

## Cdr2p is a dose-dependent regulator of cell size

To determine if the cdr2p concentration is important in cell size control, we tested the effects of varying its expression level (*Figure 3A,B*). Cdr2p was expressed from an *nmt81* promoter, regulated by thiamine in the media. Mild cdr2p overexpression in the absence of thiamine (estimated 1.6-fold) caused cells to divide at abnormally short cell lengths. Consistent with previous studies (*Breeding et al., 1998*), higher levels of overexpression caused cytokinesis defects and accumulation of longer cells. Conversely, decreased cdr2p expression led cells to divide at much longer lengths, similar to a *cdr2* null strain (*Morrell et al., 2004*; *Martin and Berthelot-Grosjean, 2009*; *Moseley et al., 2009*). Thus, cdr2p is a dose-dependent regulator of cell size and mitotic entry (*Figure 3C*). The persistence of cdr2-GFP in cells treated with the protein synthesis inhibitor cyclohexamide showed that the

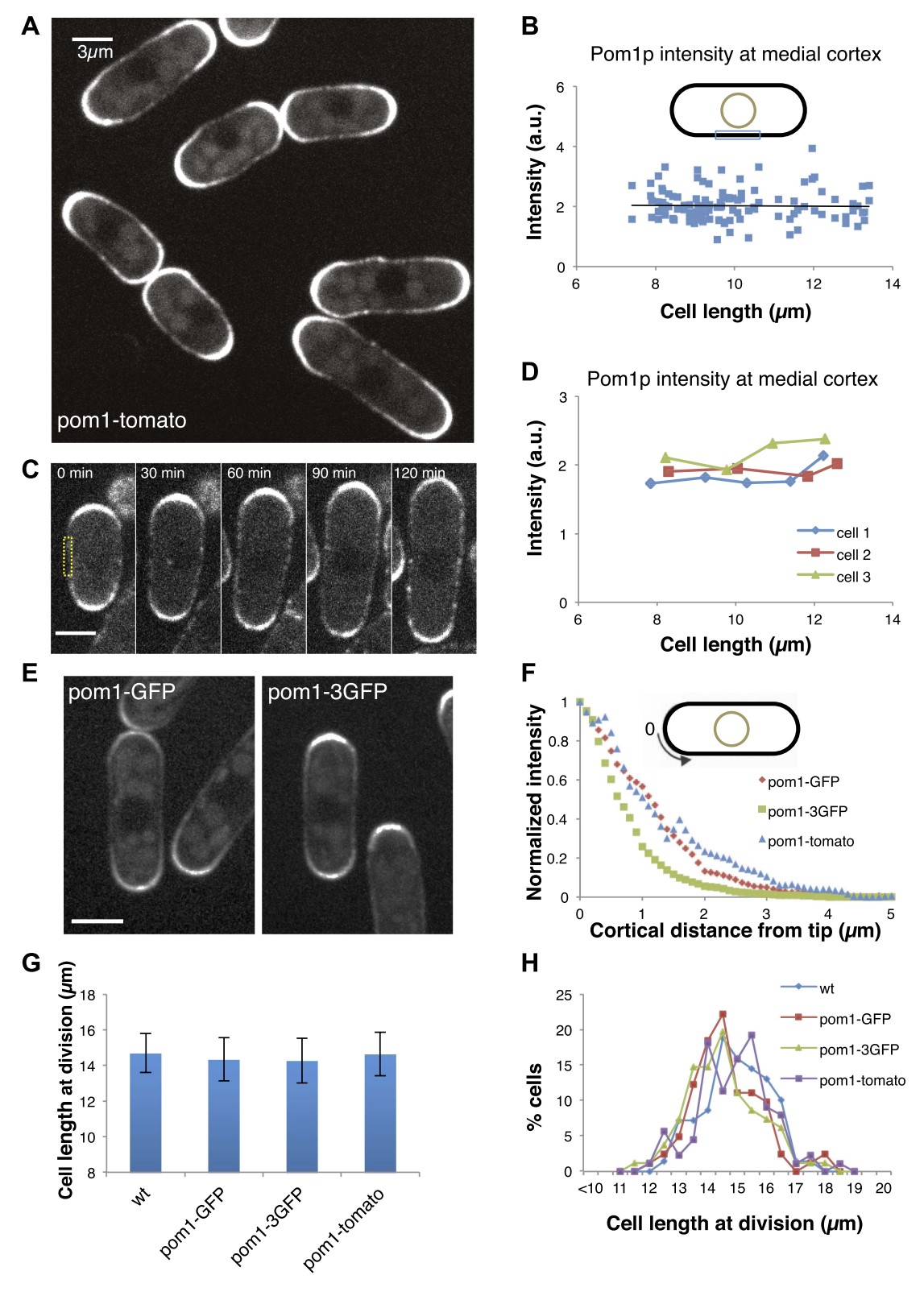

**Figure 1**. Gradient distribution of pom1p is not the basis for cell size control. (**A**) Time-averaged spinning disc confocal images of fission yeast cells expressing pom1-tomato in a medial focal plane (60 frames over 3 min). Scale bar = 3 μm. Strain used: FC2054. (**B**) Total fluorescence intensities of pom1-tomato in a medial 3-μm segment along cortical edge of interphase cells, from images like **A** (n > 100). See *Figure 1—figure supplements 1–3*.
*Figure 1. Continued on next page*

*Figure 1. Continued*

(**C**) Time-lapse images of pom1-tomato in individual cell. Images are time averaged (5 frames over 25 s) in medial focal plane. Scale bar = 3 µm. (**D**) Pom1-tomato intensities at medial cortex (as in **B**) of individual growing interphase cells. (**E**) Cells expressing pom1-GFP or the pom1-3GFP. Imaging as in **A**. Strains used: FC1162, FC2685. Scale bar = 3 µm. (**F**) Gradient profiles of pom1-3GFP, pom1-GFP and pom1-tomato (n > 30 each strain). Peak absolute protein numbers in pom1-3GFP and pom1-GFP gradients were similar. Error bars not shown for clarity. See *Figure 1—figure supplement 4*. (**G**) Effect of pom1p fusions on cell size as measured by length of septated cells (n > 100). Error bars: SDs. Strains used: FC420, FC1162, FC2685, FC2054. See *Figure 1—figure supplement 4C, 5*. (**H**) Distribution of cell lengths at division in indicated strains.

The following figure supplements are available for figure 1:

**Figure supplement 1**. Pom1p concentration at the medial cortex does not vary with cell length.

**Figure supplement 2**. Pom1p concentration at the medial cortex: Comparison with previous data.

**Figure supplement 3**. Nuclear width as a function of cell length.

**Figure supplement 4**. Pom1p gradients with different decay lengths do not affect cdr2p distribution.

**Figure supplement 5**. Pom1p gradients with different decay lengths do not affect cdr2p node intensity or number.

majority of cdr2p is highly stable in interphase cells (*Figure 3D,E*). Together, these findings suggest that as the cell grows, cdr2p is a stable protein that is synthesized to maintain a constant concentration in the whole cell, and accumulates at the medial cortex, where it promotes mitotic entry in a concentration-dependent manner (*Figure 3C*).

## Cdr2p monitors cell size not time

A critical issue in cell size regulation is whether cdr2p levels at nodes report cell size or passage of time (*Turner et al., 2012*): is cdr2p a 'sizer' or a 'timer'? To test these possibilities, we examined cdr2p behavior in cells arrested for cell growth upon treatment with an actin inhibitor Latrunculin A (*Ayscough et al., 1997*; *Chang, 1999*). Levels of a simple timer should continue to increase over time, even without cell growth, while a sizer would not increase without cell growth. Latrunculin A-treated cells exhibited no growth and no increase in cdr2-GFP levels at nodes (*Figure 4A,B*). Next, we compared cdr2p in cells growing at different rates. We used *for3Δ* (formin) mutants, which are defective in cell polarity regulation and exhibit highly variable growth rates (*Feierbach and Chang, 2001*). This mutant allowed us to measure cells in the same microscope field with identical genotype and growth conditions, but with over twofold varying growth rates (*Figure 4C,D*, *Figure 4—figure supplement 1*). The rate of cdr2-GFP accumulation at nodes strongly correlated with the rate of cell growth. (p<10⁻³ see 'Materials and methods'; *Figure 4C,D*, *Figure 4—figure supplement 1*). Thus, cdr2p has properties of a 'sizer' not a 'timer'.

## Cdr2p binds all over the cortex

Our findings raise the key question of how nodal cdr2p concentration is able to scale with cell size. In further characterizing the dynamic behavior of cdr2p, we found that in addition to cdr2p in nodes and a diffuse cytoplasmic haze, it also localized to dim, dynamic dots all around the cortex (*Figure 5A*, *Video 2*). This dim cortical population has not been described previously. Interestingly, the distribution of these dim cortical cdr2-GFP dots did not vary over the cell tip, and thus did not correlate with levels of pom1p at cell tips. Thus, cdr2p is able to bind to the whole surface of the cell.

## Mathematical models for size-dependent accumulation of Cdr2p at nodes

Because it is not intuitively clear how these dynamic behaviors of cdr2p might cause it to concentrate in the nodal region in a cell-size-dependent manner, we developed a mathematical model to probe the mechanism responsible (*Figure 5B–D*). Based on our experiments, this model postulates that: (1) the concentration of cdr2p in the cytoplasm is homogeneous and changes only slightly with cell length (*Figure 5—figure supplement 1A*); (2) cytoplasmic cdr2p molecules can bind all over the plasma membrane (*Figure 5A*; *Video 2*), and subsequently move rapidly by diffusion on the cortex;

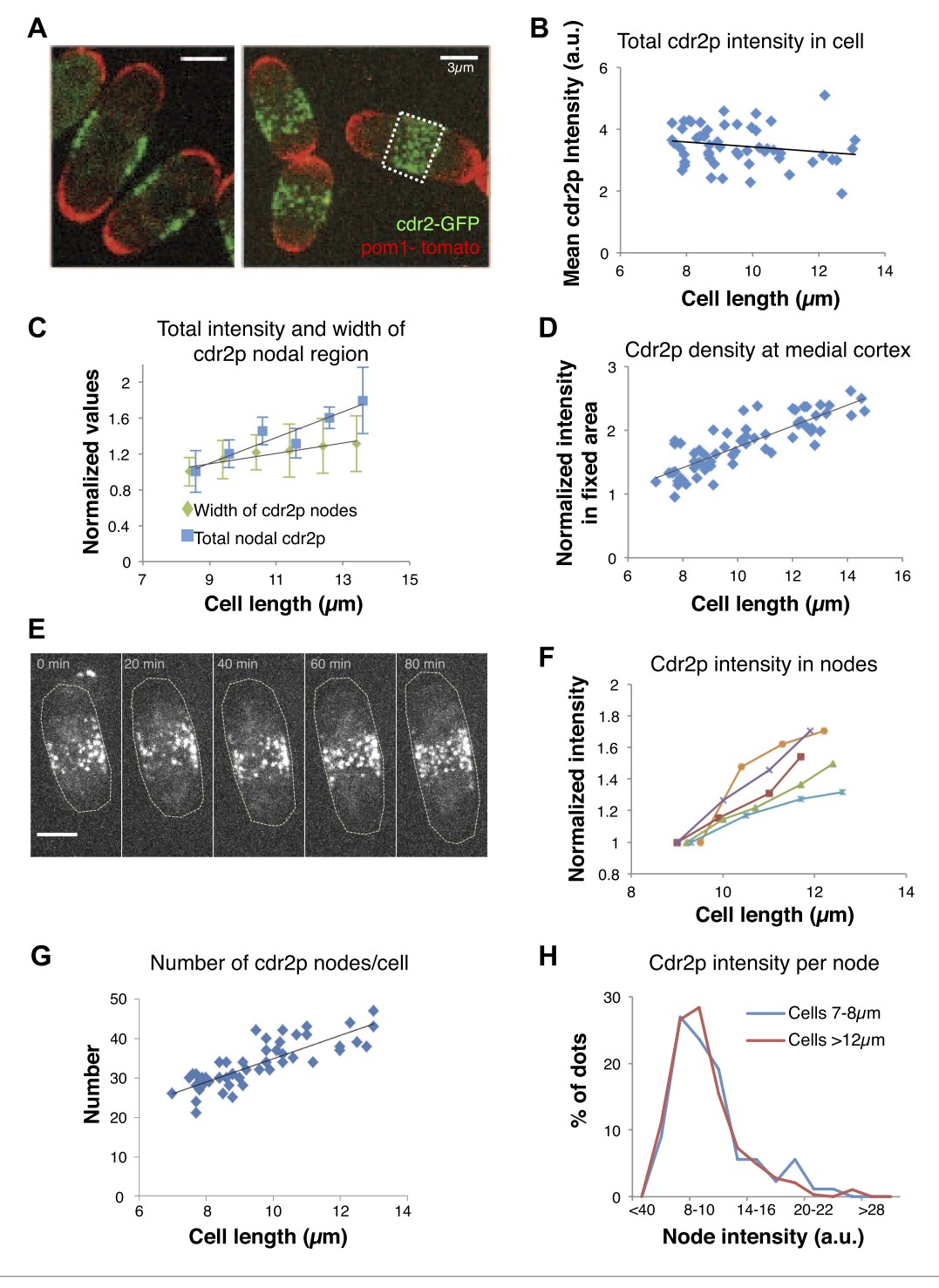

**Figure 2.** Cdr2p accumulates in nodes at the medial cortex as cells grow. (**A**) Fission yeast cells expressing pom1-tomato and cdr2-GFP. Left panel: single medial confocal section; right panel: maximum Z-projection through whole cell. Strain used: FC2678. Scale bars = 3 μm. (**B**) Total cellular intensity of cdr2-GFP in cells of different lengths. Mean intensities over the whole cell from sum projection images. n = 54 cells. Black line: linear fit with r² = 0.04. (**C**) Cdr2-GFP total intensity in medial cortex (blue) (*Figure 2—figure supplement 3A*; n = 51) and width of cdr2-GFP nodal region along long cell axis (green) as function of cell length (n = 185). Error bars = SEM. Black lines: linear fits, r² = 0.90 and 0.89 for width and intensity respectively. See *Figure 2—figure supplements 1–4*. (**D**) Total cdr2-GFP intensity in fixed a 3-μm wide medial band as function of cell length (*Figure 2—figure supplement 3F*) *Figure 2. Continued on next page*

*Figure 2. Continued*

n = 67. Black line: linear fit with r² = 0.71. (**E**) Time-lapse maximum projection images of a cell expressing cdr2-GFP. Scale bar = 3 μm. (**F**) Total normalized intensities of nodal cdr2-GFP in 5 cells tracked over time (measured from images like **E**, using method of *Figure 2—figure supplement 3A*). See *Figure 2—figure supplement 5*. (**G**) Number of cdr2-GFP nodes as function of cell length (n = 51). Black line: linear fit with r² = 0.67. Nodes identified by thresholding, using method of *Figure 2—figure supplement 3C*, which provides a lower-bound estimate. (**H**) Distributions of cdr2-GFP node intensities in short vs long cells. n = 89 nodes in 9 cells, n = 286 nodes in 7 cells, respectively (nodes as determined in **G**).

The following figure supplements are available for figure 2:

**Figure supplement 1**. Measurement of cdr2p protein number.

**Figure supplement 2**. Cdr2p and pom1p intensity measurements as a function of cell length.

**Figure supplement 3**. Comparison of image analysis methods for quantitating cdr2p fluorescence in the nodes.

**Figure supplement 4**. FRAP analysis of cdr2-GFP.

**Figure supplement 5**. Cdr2p node number but not maximal intensity in each node increases with cell length.

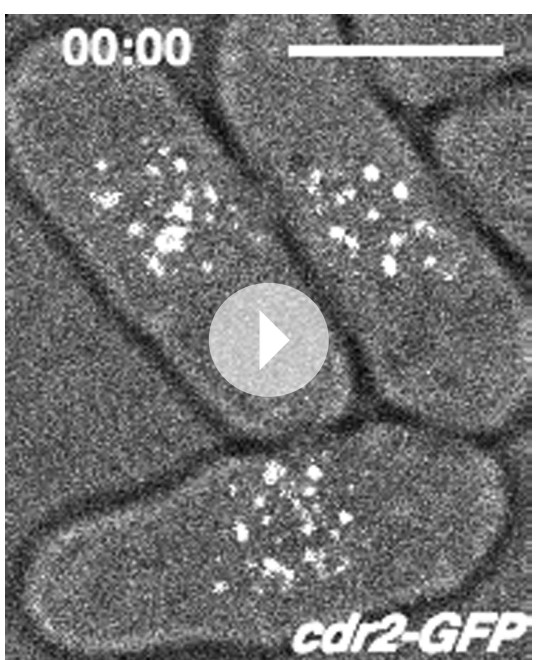

**Video 1**. Cdr2-GFP nodes in wild-type fission yeast cells. Fission yeast cells expressing cdr2-GFP. Spinning disc confocal images in a cortical slice, acquired every 20 s. Initial images show a brightfield/fluorescence image to show the cell outline. Scale bar: 5 μm. Strain FC2688. Time stamp = min, sec.

(3) cortical cdr2p molecules can transition to associate with a nodal region on the medial cortex. Note that the details of the formation and growth of individual nodes are beyond the scope of the model. Rather, we simply model the overall number of cdr2p molecules in the nodal region. (4) Both cortical and nodal cdr2p can then unbind and return cdr2p to the cytoplasm; (5) cytoplasmic cdr2p can then diffuse rapidly before rebinding to the membrane. As the timescale of cell growth (hours) is much slower than the timescale of the cdr2p dynamics (minutes, *Figure 2—figure supplement 4*), we assumed that the molecular system is, at any given time, effectively in steady state. This steady-state assumption is also consistent with experimental findings that cdr2p levels at nodes are stable over time when cells are not growing (*Figure 4A,B*). This model (*Figure 5B*) was implemented by two mass-action equations for cortical and nodal cdr2p, and solved analytically (*Figure 5C*).

Importantly, the model reveals how the cdr2p dynamics ensure a nodal cdr2p density that scales with cell size, or more specifically, with the surface area of the plasma membrane (*Figure 5C*). The simplicity of the model allowed us to clarify the two key elements important for this area sensing. The first is that the area of the nodal region must not scale proportionally with the total cell membrane area as the cell size increases (*Figure 2C*). We then have one process (cdr2p membrane association) that scales proportionally with cell area, with a second process (uptake of cortical cdr2p into the nodes), which does not. The second key element is that the nodal region receives information via cdr2p about the entire surface area of the cell. In this model, cdr2p needs to be able bind the membrane long enough to move on the membrane to reach the nodal region. The outcome is then a rising

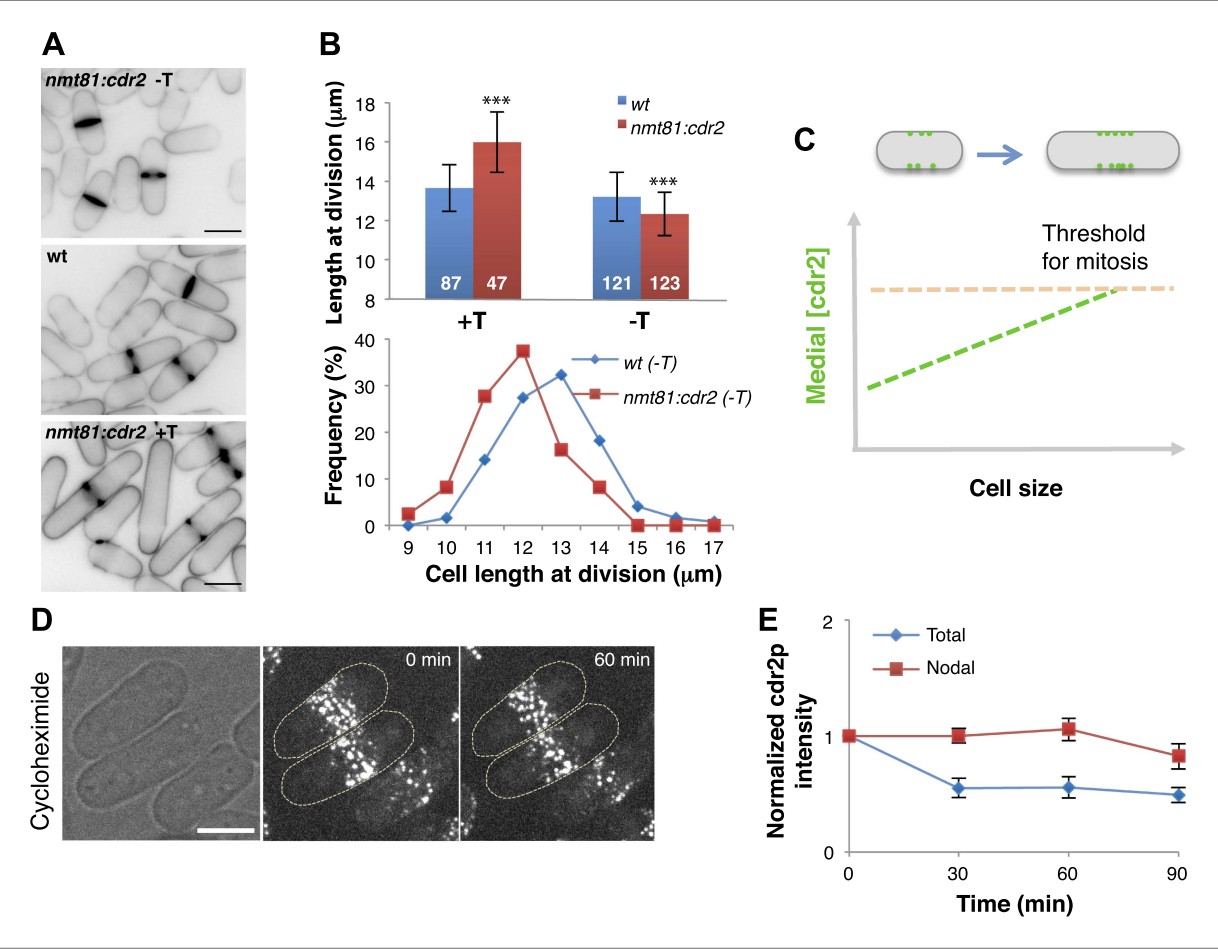

**Figure 3**. Cdr2p is a dose-dependent regulator of cell size. (**A**) Effect of cdr2p expression level on cell size. *cdr2⁺* was expressed at different levels using a thiamine-regulatable promoter (*nmt81-cdr2*). Inverted images of cells stained with cell wall dye blankofluor. Cells express cdr2p at levels on average of 1.6, 1.0 and 0.3-fold relative to wild type (top to bottom). Strains used: FC15, FC2691. Scale bar = 5 μm. (**B**) Length of cells at septation. n = 87, 47, 121, 123 cells. T = thiamine. Error bars = SD, ***p<0.0001 as determined by Kolmogorov–Smirnov statistical tests. (**C**) Model that the local concentration of cdr2p increases in the region of the medial cortical nodes as cells grow, and when it reaches a critical level, promotes entry into mitosis. (**D**) Stability of cdr2-GFP protein. Time-lapse images of cells expressing cdr2-GFP treated with 100 μg/ml cycloheximide (***Polanshek, 1977***). Strain used: FC2688. Scale bar = 5 μm. (**E**) Total cell and nodal cdr2-GFP intensities in individual cells treated with cycloheximide over time. Total cell intensity was measured as in ***Figure 2B*** n = 9 cells. Error bars = SD.

cdr2p nodal density with increasing cell area. Using the experimentally determined nodal/cortical areas, and with other parameters measured/constrained from our experiments (***Figure 5D***, 'Materials and methods'), we fitted the cdr2p density in the medial nodes as a function of cell length to that measured experimentally, with good results (***Figure 5E***). Note that wild-type *S. pombe* cells are rod-shaped and have an approximately constant width, so that surface area and cell length are proportional to one another. A more sophisticated version of the same underlying model, including spatially varying cdr2p on the cortex, generated similar results (***Figure 5—figure supplement 1B–F***, 'Materials and methods').

In addition, alternative models in which cdr2p does not need to diffuse long distances on the plasma membrane to the nodes are also consistent with the current findings. We analyzed a model in which cdr2p was now modified (e.g., phosphorylated) at the cortex and remains modified for a period even if it returns into the cytoplasm, from where it then can diffuse to and accumulate at the nodal region (***Figure 5—figure supplement 2A***). The underlying area-sensing mechanism was nevertheless conserved in this alternative model, with similar key elements as discussed above (***Figure 5—figure supplement 2B***, 'Materials and methods').

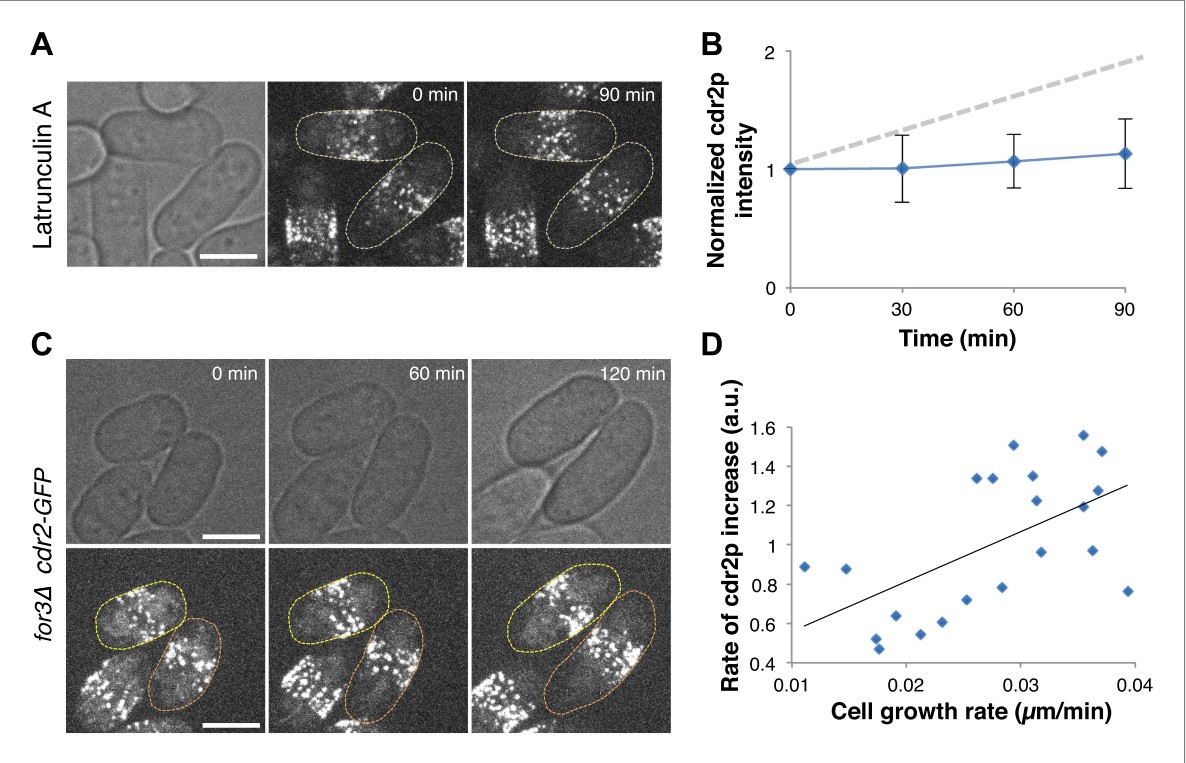

**Figure 4**. Cdr2p has properties of a sizer but not a timer. (**A**) Cdr2-GFP does not accumulate at nodes over time without cell growth. Time lapse images of cells in which growth was halted by 200 μM Latrunculin A, an actin inhibitor (*Chang, 1999*). Scale bar = 5 μm. Strain used: FC2688. (**B**) Mean cdr2-GFP nodal intensity over time in individual LatA-treated cells. n = 9 cells. Error bars = SD. Dotted line shows for comparison the observed average increase of nodal cdr2-GFP in untreated, growing cells (from *Figure 2F*). (**C**) Time-lapse images of cdr2-GFP in *for3Δ* (formin) cells. Two sister cells which exhibit variable growth rates are highlighted (yellow and orange outlines). Strain used: FC2690. Scale bar = 5 μm. (**D**) Correlation between cell growth rate and the rate of accumulation of cdr2-GFP in *for3Δ* cells (*Figure 4—figure supplement 1*). Line is linear fit to the data (r² = 0.34). A 'timer' molecule would show no correlation. n = 21 cells.

The following figure supplements are available for figure 4:

**Figure supplement 1**. Rate of cdr2p nodal accumulation correlates with the rate of cell growth.

## Cdr2p scales with cell surface area not volume

A central prediction of the modeling is that cdr2p is sensing the surface area of the cell: cdr2p at nodes should scale with surface area, and not, for instance, cell volume. To experimentally test if cdr2p scales with surface area or volume, we analyzed cdr2p levels in *S. pombe* mutants with different widths, so that surface area and volume are uncoupled. Rga2p and rga4p are Rho-GAPs involved in regulation of cell polarity and width (*Das et al., 2007*; *Villar-Tajadura et al., 2008*; *Kelly and Nurse, 2011*); *rga2Δ* mutants are thinner while *rga4Δ* mutants are fatter than wild type (*Figure 6A*). We measured surface areas and volumes in these cells ('Materials and methods'; *Figure 6—figure supplement 1*). In a group of interphase cells of similar surface area but of different volumes, nodal cdr2-GFP intensities correlated with surface area (*Figure 6B*). Conversely, in considering cells of similar volume but with a range of different surface areas, cdr2-GFP nodal intensity correlated with surface area and not volume (*Figure 6C*). These results thus suggest that nodal cdr2p scales with cell surface area, in agreement with the predictions of the mathematical models.

## Cells enter mitosis at a given surface area

These findings lead to another key prediction that cells enter mitosis at a specific cell surface area. We measured cell length, surface area and volume in wild-type, *rga2Δ* and *rga4Δ* strains in dividing cells; these dimensions are indicative of the size of the cells at entry into mitosis. These cells with different shapes entered mitosis with more similar cell surface areas but differing cell volumes and lengths

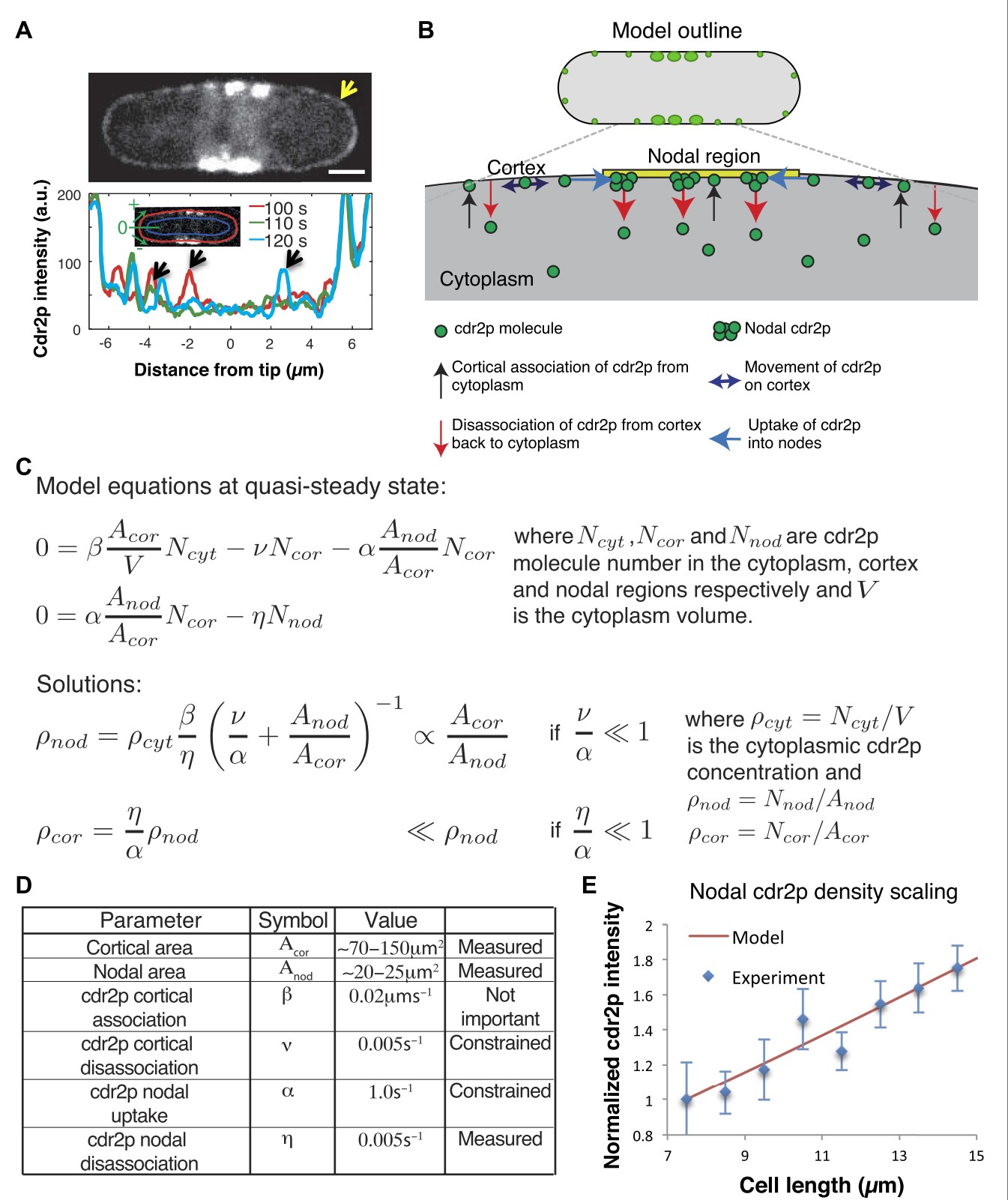

**Figure 5**. Model for cell size-sensing by cdr2p. (**A**) Confocal time-averaged image (60 frames over 10 min) in medial focal plane of cell expressing cdr2-GFP. Arrow highlights dim cdr2-GFP all around cell cortex (*Video 2*). Scale bar = 2 μm. Graph shows cdr2-GFP profiles on cortex around one pole at indicated time points. Cdr2-GFP appears brighter in the cytoplasm around nodes due to out-of-focus nodal fluorescence. Black arrows
*Figure 5. Continued on next page*

*Figure 5. Continued*

denote local peaks in the cdr2-GFP signal that are clearly distinct from the mean cdr2-GFP cortical signal. Strain: FC2678. (**B**) Outline of mathematical model for cdr2p dynamics. (**C**) Equations and analytic solutions describing cortical and nodal cdr2p number. (**D**) Model parameters. 'Measured': deduced directly from experiment, 'constrained': limited by nodal cdr2p density scaling with cell length, 'not important': plays no role in nodal cdr2p density scaling. (**E**) Model fit to nodal cdr2-GFP density as function of cell length (data from maximum intensity projection as described in *Figure 2—figure supplement 3A*, with cells binned by length at 1 µm intervals). Equations and parameters given in **C**, **D**. Error bars = SD. See *Figure 5—figure supplement 1, 2*.

The following figure supplements are available for figure 5:

**Figure supplement 1**. Spatial membrane model for cdr2p distribution.

**Figure supplement 2**. Cdr2p-modification model.

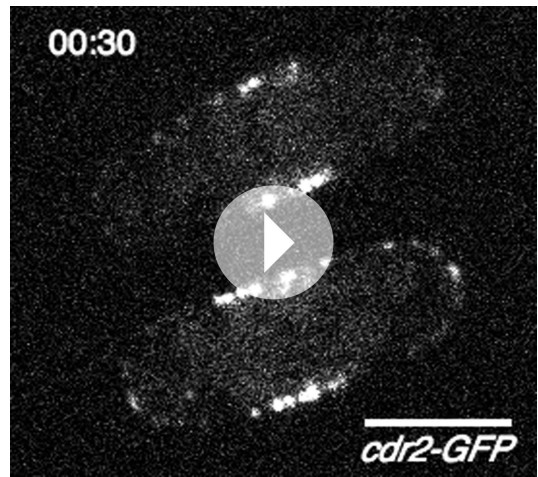

**Video 2**. Dynamics of cortical cdr2p in wild-type cells. Fission yeast cells expressing cdr2-GFP. Spinning disc confocal images in a medial slice, acquired every 10 s over 5 min. These images, which were taken at longer exposures and higher laser power than the other two videos, reveal dim dynamic cdr2p dots all over the cortex. Scale bar: 5 µm. Strain FC2688. Time stamp = min, sec.

(*Figure 6D,E*). All three strains exhibited average surface areas of 150 µm² ± 8 µm², while the average volumes varied from 120 µm³ to 150 µm³ and average lengths from 11 µm to 15 µm. A more rigorous analysis based on Jensen–Shannon distances ('Materials and methods') showed quantitatively that the distributions of surface area were more similar than those for volume or length. These findings suggest that cells monitor their size at the G2/M transition by measuring their surface area.

## Regulation of cdr2p nodes by pom1p

We next examined how pom1p quantitatively affects cdr2p. In *pom1Δ* mutants, cdr2p is thought to be somehow more 'active' and promotes division at slightly shorter cell lengths than wild type (*Martin and Berthelot-Grosjean, 2009*; *Moseley et al., 2009*). In *pom1Δ* cells, cdr2-GFP is spread in dots throughout much of the cortex, except for the growing cell tip (*Figure 7A*; *Celton-Morizur et al., 2006*; *Padte et al., 2006*; *Martin and Berthelot-Grosjean, 2009*; *Moseley et al., 2009*). The total amount of cdr2p in the cell was similar in wild-type and *pom1Δ* mutant cells over a range of cell lengths (*Figure 7B*). Cortical profiles showed that in *pom1Δ* cells, cdr2p was still enriched over the medial cortex and that the non-growing end had levels roughly half that of the medial region (*Figure 7C*). The fraction of cdr2p that is cortical and the area of nodal cdr2p were both substantially increased in *pom1Δ* cells (*Figure 7D,E*). Interestingly, the increase of cortical cdr2p with cell length was similar in *pom1Δ* vs wild-type cells, but the curve was shifted slightly upwards (*Figure 7F*). In contrast, in the medial cortical region, cdr2p levels were lower than wildtype (*Figure 7G*). A simple interpretation is that cdr2p is able to signal to promote entry into mitosis from nodes on non-medial sites in this mutant. However, another factor to consider is that cdr2p kinase activity may also be altered in these cells. Time-lapse imaging showed that cdr2p nodes are more motile in *pom1Δ* cells than in WT (*Figure 7H*; *Videos 1, 3*), suggesting a defect in the anchoring of these nodes in the membrane. At the growing end, there are also the dim cortical motile cdr2p dots, similar to those present at cell ends in WT (*Figure 5A*). Thus, pom1p affects the distribution and mobility of cdr2p nodes.

We also examined the effect of disrupting pom1p localization on the cdr2p distribution. A construct in which pom1p is targeted all over the plasma membrane has been described (PMT-Pom1C fusion, *Figure 7—figure supplement 1A*) (*Moseley et al., 2009*). Although cdr2p was expressed at

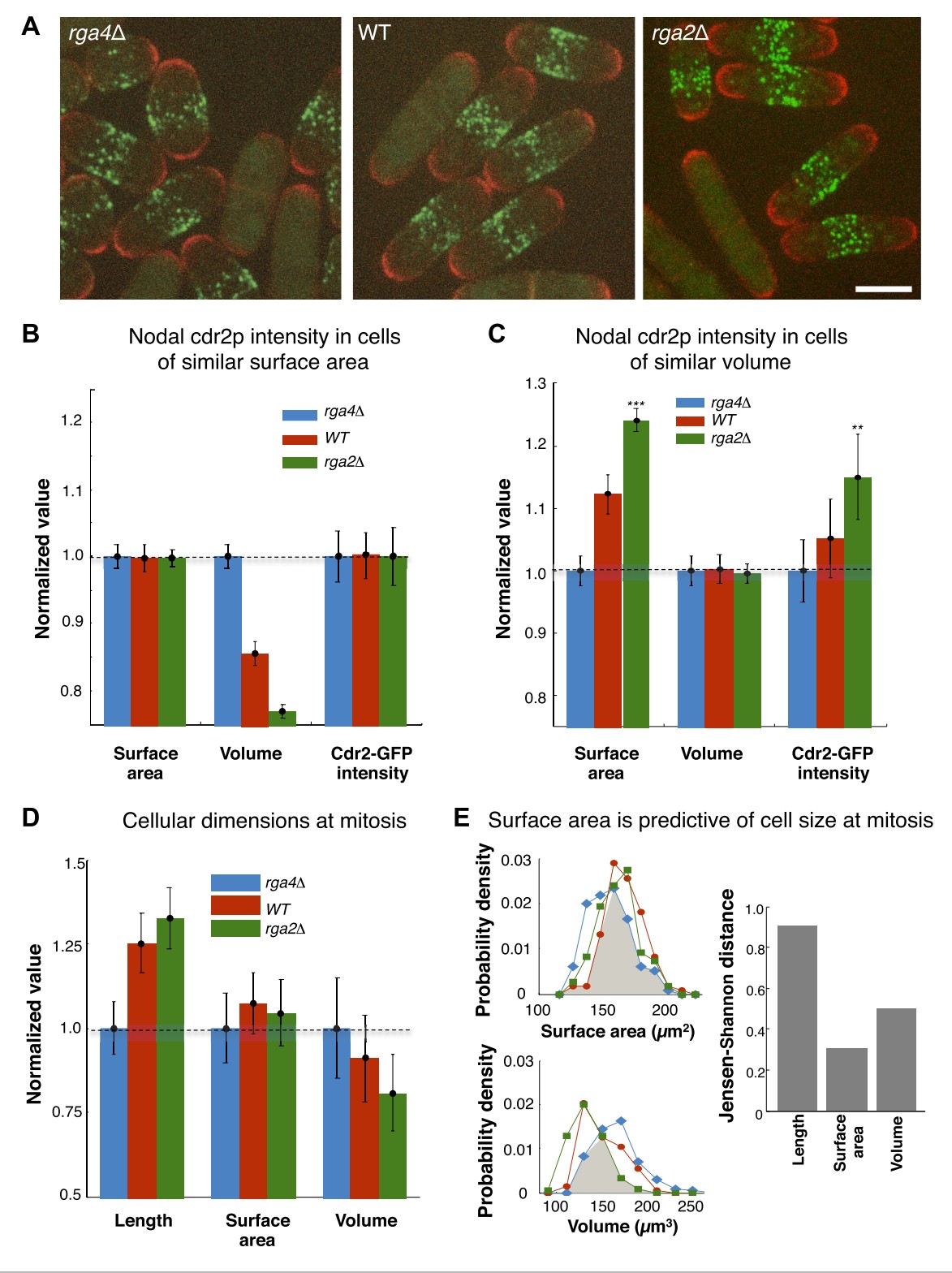

**Figure 6**. Cdr2p and cell size at division scale with cell surface area. (**A**) Fission yeast cells expressing cdr2-GFP and pom1-tomato in wt, *rga4Δ* (fat morphology) and *rga2Δ* (thin morphology) backgrounds. Maximum Z-projection images. Cells lacking nodes are in mitosis. Strains used: FC2678, FC2794, FC2795. Scale bar = 5 μm. (**B**) Comparison of measured nodal Cdr2-GFP intensity in cells of different volumes. For each cell, the surface area and volume were measured by segmentation ('Materials and methods'). A subset of cells whose surface area was within 10–20% of the mean surface area was selected for each cell type ('Materials and methods'). The graphs show the surface area, volume, and nodal cdr2-GFP intensity (cdr2-GFP intensity

*Figure 6. Continued on next page*

*Figure 6. Continued*

measured as defined in *Figure 2—figure supplement 3A*) in these selected cells. For each data type, normalization is by mean value for *rga4Δ* cells. Error bars = Error on the mean. n = 24 (wt) cells, 27 (*rga4Δ*), 32 (*rga2Δ*). Strains used in **B** and **C**: FC1441, FC2792, FC2793. See *Figure 6—figure supplement 1*. (**C**) As in **B**, except groups of cells were selected with similar volumes (mean measured volume ± 10–20%). n = 24 (wt) cells, 27 (*rga4Δ*), 27 (*rga2Δ*). These data show cdr2-GFP scaling with surface area. The difference in surface area and cdr2-GFP intensity between the *rga2Δ* and *rga4Δ* cells is statistically significant (\*\*p<10$^{-3}$, \*\*\*p<10$^{-4}$). See *Figure 6—figure supplement 1*. (**D**) Comparison of cell lengths, surface areas and volumes in *rga4Δ*, wild type and *rga2Δ* at time of septation ('Materials and methods'). The septum is not included in these measurements. Data for each set is normalized by the appropriate value for the *rga4Δ* cells. Error bars = SD. Strains used: FC2554, FC2555, FC2556. n = 76 (wt), 64 (*rga4Δ*), 60 (*rga2Δ*). (**E**) Quantitating differences between *rga4Δ*, wt and *rga2Δ* at time of septation. Left: probability density distributions for measured surface area (top) and volume (bottom) for wild type (red), *rga2Δ* (green) and *rga4Δ* (blue) cells in (**D**). Gray area marks the overlap region between the distributions. Error bars not shown for clarity. Right: to quantitatively compare these distributions, we calculated the Jensen–Shannon distance (*Lin, 1991*) between the length, surface area and volume distributions for the different cell types (where 1 corresponds to the distributions having no shared information and 0 to identical distributions, see 'Materials and methods'). This analysis shows that these cells with different shapes divide with similar surface area.

The following figure supplements are available for figure 6:

**Figure supplement 1**. Scaling of nodal cdr2-GFP intensity with surface area and volume.

normal levels in the whole cell, it was evenly distributed all over the cortex at a low level, and did not increase at the medial cortex with increasing cell length (*Figure 7*, *Figure 7—figure supplement 1*). As shown previously (*Moseley et al., 2009*), these cells divided at abnormally long cell lengths, similar to *cdr2Δ* mutants. These data show that pom1p has an inhibitory effect on cdr2p localization to nodes. These results further provide support that cdr2p needs to be present at these medial nodes in order to function effectively in cell size control.

## Discussion

Here we propose a mechanism for cell size sensing based on a cortical sizer protein cdr2p. We provide evidence that cells sense a critical cell size by measuring cell surface area rather than, for example, cell volume or absolute length, a mechanism that could function regardless of the cell shape. As the cell grows, the concentration of cdr2p at the medial cortex increases. We have developed models explaining how cdr2p probes the surface area of the cell, and conveys this information to the medial cortex. There, cdr2p may signal to cell cycle regulators located on the nearby spindle pole body and nucleus (see below). When the cell reaches a critical size, cdr2p at the nodes may reach a critical local concentration that promotes mitotic entry.

Our quantitative models show how cdr2p can convey information about global cell area and deliver it in the form of a local (nodal) concentration. This size-sensing model shares elements with a proposed microtubule length control mechanism termed the 'antenna model'. In the microtubule model, longer microtubules bind more motor proteins, which then accumulate at the microtubule end in a length-dependent manner (*Varga et al., 2006*). In the cell size sensing case, the whole surface area of the plasma membrane may be regarded as an 'antenna'. Similar to the microtubule model, the property of cdr2p to first bind to the plasma membrane 'antenna' (as opposed to merely binding the nodes directly) is critical for this mechanism to read out cell size. This membrane cdr2p must then transit to the nodal region, where the cdr2p nodal density serves as a read-out of cell area. Although cdr2p may not exhibit directed motor-driven movements, this movement can still occur by diffusion along the membrane. We also considered an alternative model, where cdr2p is modified on the membrane, but after unbinding is able to diffuse through the cytoplasm to the nodes. The modification allows information about membrane area to be preserved in the cytoplasm, from where it can be relayed to the nodes (*Figure 5—figure supplement 2A*). Furthermore, as the amount of nodal cdr2p reflects cell size rather than time, we postulate that the system is effectively in a dynamic steady state at a given cell size, with fast cdr2p dynamics compared to the timescales of cell growth.

The localization of a cdr2p sizer at cortical nodes provides several key advantages over other locales. First, it allows the local concentration of nodal cdr2p to increase as the cell grows. Previously proposed mechanisms have been based upon nuclear concentration or the nuclear/cytoplasmic ratio of a sizer, but in many cell types (including fission yeast), nuclear volume also increases as cells grow (*Neumann and Nurse, 2007*; *Figure 1—figure supplement 3*). Second, we speculate that

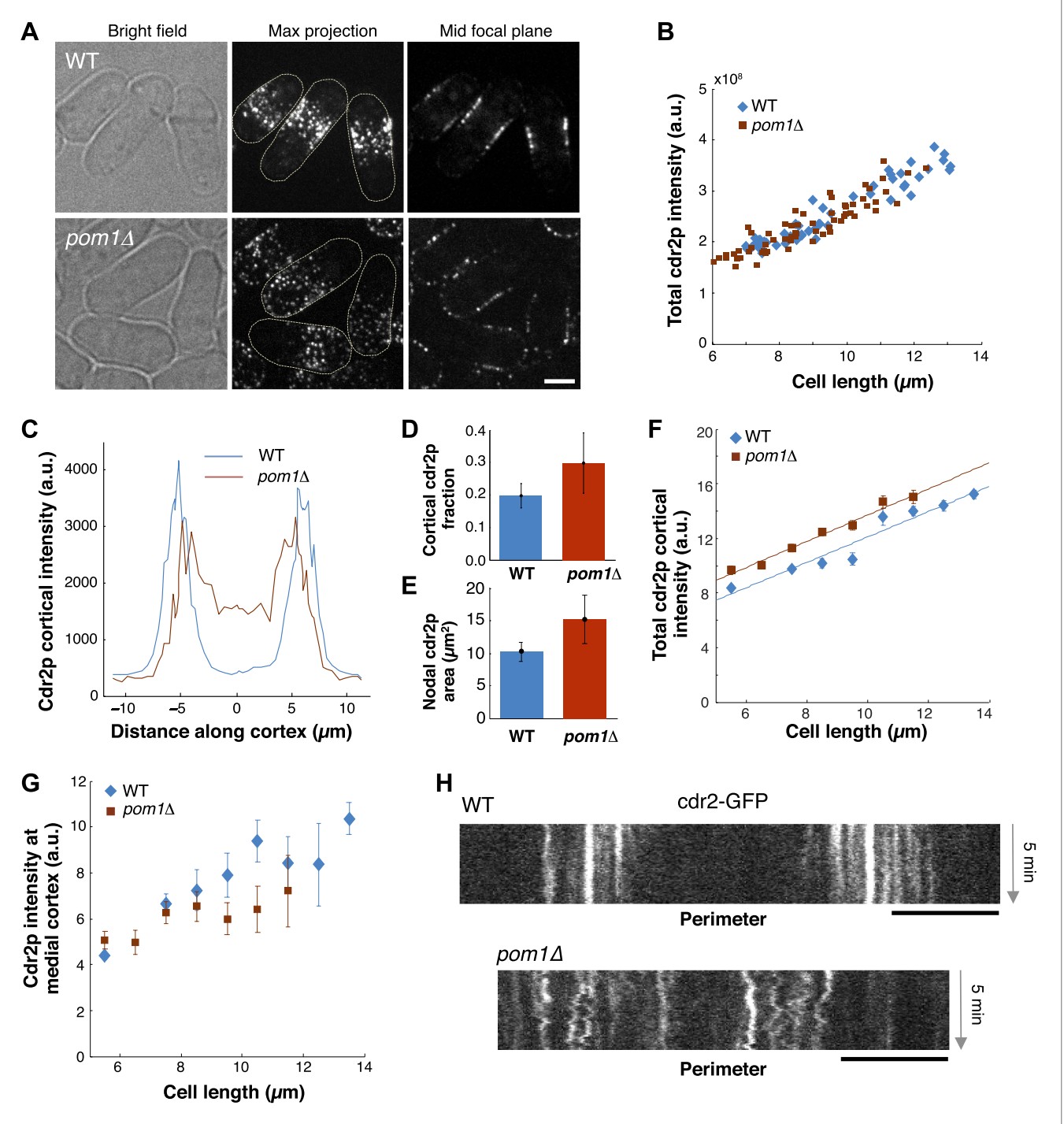

**Figure 7**. Cdr2p behavior in *pom1Δ* mutants. (**A**) Fission yeast cells expressing Cdr2-GFP in wt and *pom1Δ* background. Brightfield, maximum projection, and mid–focal plane images are shown. Strains used: FC1441 and FC2057. Scale bar = 3 μm. (**B**) Comparison of total measured cdr2-GFP intensity (from sum projection after background subtraction) with cell length. n = 52 (wt), 72 (*pom1Δ*). (**C**) Average cdr2-GFP intensity profile around cortex of cell (spatial direction as defined in cartoon in *Figure 1F*). *pom1Δ* cells are orientated such that the cell end with the higher cdr2-GFP level is defined to be at d = 0 μm. n = 52 (wt) cells, 72 (*pom1Δ*). Error bars not shown for clarity. See 'Materials and methods' for further details. (**D**) Fraction of cdr2-GFP signal observed on the cortex compared with total measured cdr2-GFP in the medial plane. The cortical signal is calculated as the sum of measured intensity along a mask around the cortex (see 'Materials and methods' for mask definition). The total signal is defined as the total measured cdr2-GFP intensity on and inside the mask. Error bars = SD. n = 52 cells (wt), 72 (*pom1Δ*). (**E**) Measured area of nodal cdr2-GFP region from maximum intensity projection images. Regions were measured manually for individual cells. n = 46 (wt) cells, 77 (*pom1Δ*). Error bars = SD. (**F**) Accumulation of total membrane

*Figure 7. Continued on next page*

*Figure 7. Continued*

cdr2-GFP (both nodal and cortical signal) against cell length. n = 52 (wt) cells, 72 (*pom1Δ*). See 'Materials and methods' for details. Lines are linear least-square fits to the data, with similar slopes. Error bars = SD. (**G**) Accumulation of nodal cdr2-GFP (maximum intensity projection) within 3 µm medial cortical region. n = 52 cells (wt), 72 (*pom1Δ*). Error bars = SD. (**H**) Kymograph of cortical cdr2-GFP over 5-min period in wild type and *pom1Δ* cells. Scale bars = 5 µm.

The following figure supplements are available for figure 7:

**Figure supplement 1**. Cdr2p behavior in cells in which pom1p is targeted all over the cortex.

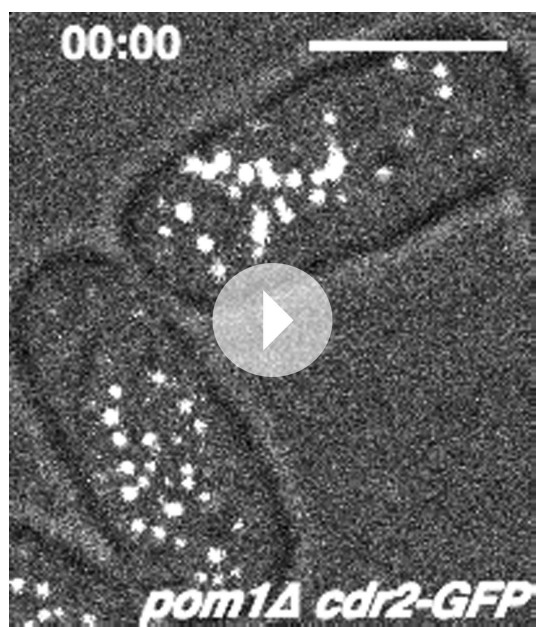

**Video 3**. Abnormal behavior of cdr2p nodes in pom1Δ cells. Fission yeast cells expressing cdr2-GFP. Spinning disc confocal images in a cortical slice, acquired every 20 s. Scale bar: 5 µm. Strain FC2057. Note that many of these nodes appear more motile than those in wildtype cells (***Video 1***). Time stamp = min, sec.

medial cortical placement of nodes surrounding the medial nucleus may allow cdr2p to communicate its local concentration to presumed targets such as wee1p and cdk1p on the nucleus. Although wee1p can be observed at some nodes upon overexpression (***Moseley et al., 2009***), its localization in late G2-phase is clear in the nucleus, and at the spindle pole body (SPB) (***Masuda et al., 2011***), a structure on the nuclear envelope situated close (often <0.5 µm) to the nodes. Cdk1/cyclin B and polo kinase are also located at the SPB and nucleus (***Alfa et al., 1990***; ***Masuda et al., 2011***; ***Grallert et al., 2013***). Potentially, the SPB could detect local gradients of cdr2p (or other molecules) emanating from nearby cortical nodes. However, as a simple cdr2p concentration gradient in the cytoplasm is expected to be very shallow (due to rapid diffusion), it is likely that additional layers of regulation such as through phosphorylation states or diffusion barriers would be needed to generate suitably steep gradients. The potential importance of the geometric relationship between the nodes and SPB/nucleus remains to be tested.

The localization of these nodes to the medial cortical region involves multiple inputs. One important contributor is pom1p. Although pom1p clearly regulates cdr2p function and phosphorylation, our data indicate that the pom1p gradient distribution may not be the primary size sensing mechanism as previously proposed. Indeed, our data are consistent with a recent report that size correction still occurs in *pom1Δ* mutants (***Wood and Nurse, 2013***). Rather, a primary role of pom1p may be to ensure the medial localization of nodes. Thus, pom1p may affect cdr2p nodes in part by affecting distribution and general properties (such as its mobility in the membrane) of the nodes. Recent studies (published while this work was in press) suggest that cdr2p activity is also regulated by phosphorylation by pom1p and ssp1p protein kinases (***Bhatia et al., 2014***; ***Deng et al., 2014***). Another important factor in cdr2p localization is likely to be the nucleus that is situated in the cell interior with roughly the same width as the nodal region. Studies on mid1p, another component of the nodes, suggest that the nucleus governs dynamic nodal localization, in a mechanism that may involve nuclear shuttling (***Paoletti and Chang, 2000***; ***Daga and Chang, 2005***; ***Almonacid et al., 2009***). Furthermore, the organization of the cortical endoplasmic reticulum also influences nodal stability and localization (***Zhang et al., 2010***). There are also likely to be additional (or alternative) inputs into size control (***Coudreuse and Nurse, 2010***; ***Navarro and Nurse, 2012***; ***Wood and Nurse, 2013***). Additional cell size regulators include the cell tip protein nif1p (***Martin and Berthelot-Grosjean, 2009***; ***Wood and Nurse, 2013***), and skb1p, which localizes to cortical patches distinct from the nodes (***Deng and Moseley, 2013***).

Cells expressing a cdk1p–cyclinB fusion still exhibit apparently near-normal size control in the absence of wee1p/mik1p or cdk1p-tyrosine phosphorylation control (*Coudreuse and Nurse, 2010*; *Navarro and Nurse, 2012*; *Wood and Nurse, 2013*), suggesting the existence of controls that are entirely independent of cdk1–tyrosine phosphorylation. Thus, this simple cdr2p-based mechanism is likely to be a core component of a larger network responsible for cell size control.

## Materials and methods

### *S. pombe* strain construction

Standard methods for *S. pombe* growth and genetics were used (*Moreno et al., 1991*). In general, strains were constructed using a PCR-based homologous recombination method to insert markers in the yeast chromosome (*Bahler et al., 1998*). Pom1-mGFP (= pom1-GFP) and pom1-3GFP strains were constructed by inserting mGFP and 3GFP constructs into the *pom1+* chromosomal locus from fragments amplified from pFA6a-mGFP-kanMX6 (monomeric GFP A206K [*Zacharias et al., 2002*]) and pFA6a-3GFP-kanMX6 (triple tandem GFP) (*Wu and Pollard, 2005*; *Martin and Chang, 2006*), (from JQ Wu). In general, constructs were checked by PCR and sequencing, and strains were outcrossed multiple times.

### Imaging *S. pombe* cells

For live cell imaging, *S. pombe* cells were typically grown in exponential phase in liquid YE5S media at 25°C with shaking for 18–24 hr. In some experiments, the cells were mounted in liquid YE5S media directly on glass. For long term imaging experiments, the cells were placed in open 35-mm glass bottom dishes (MatTek Corp, Ashland, MA). To stick cells to the glass, dishes were coated with lectin by drying 5 µl of 1 µg/µl lectin on the dishes; the cells in media were applied and incubated for 5 min and then 2 ml YE5S were added (*Figures 1C,D, 2E,F*). The cells were also imaged on 1% agarose YE5S pads under a glass coverslip.

### Modulating cdr2p expression levels

For experiments to alter *cdr2* levels (*Figure 3*), the *nmt81* promoter (*Basi et al., 1993*) was inserted upstream of the *cdr2* chromosomal locus by homologous recombination using a PCR-generated DNA fragment derived from *pFA6a-kanMX6-P81nmt1* (*Bahler et al., 1998*) using the following primers:

   nmt-cdr2-F: (5′-TATGCTGTTCTATGAATGGGGTTTGGATTTGGCCATCACCACTTCACCGATTT ACTGGTTCTTTTGAATAGTTGAAGTGTGAATTCGAGCTCGTTTAAAC-3′)

   nmt-cdr2-R: (5′-TTGGCTAAACGTGATGAATTTGGTCCTCCTGATCCTAAGGAAAGACCAAGC TCCCAAGGTCCAACTTCTGAAATTGTACTCATGATTTAACAAAGCGACTATA-3′).

   Correct insertion was verified by PCR of both sides of the construction using specific primers for the endogenous and inserted DNA. Multiple transformants showed the same cell size phenotypes. *nmt81-cdr2* cells and the parental wild-type strain (FC15) were grown in EMM +5 µg/ml thiamine at 25°C for 2 days, keeping the $OD_{600}$ of the culture below 0.5 over the entire period. The cells were then washed three times by centrifugation at 2000 rpm with EMM, innoculated into EMM with or without 5 µg/ml thiamine, and then grown with shaking at 25°C for 20 hr, and then samples were collected for microscopy for cell length measurements and for RNA preparation. Relative RNA expression levels were assayed by RT-PCR. RNA was isolated from cells using the RNeasy Mini Kit (Qiagen, Germantown, MD). 50 ng of RNA was used for real-time PCR using iScript One-Step RT-PCR kit with SYBR green (Bio-Rad) on a Bio-Rad Real-Time PCR system. The actin *act1* mRNA was used as standard. Amplicons for *act1* or *cdr2* were generated with the following primers:

   act1-F (5′-GAAGAAGAAATCGCAGCGTTGG-3′), act1-R (5′-CGCTTGCTTTGAGCTTCATCAC-3′)
   cdr2-F (5′-TGGGAGCTTGGTCTTTCCTTAG-3′), cdr2-R (5′-TAGCCTGTTGGCTCGAAGTAAG-3′).

   Expression levels of *cdr2* in *nmt81-cdr2 cells* were measured as fold change relative to levels in a wild-type (FC15) strain grown under the same conditions. The changes in cell lengths were consistent in multiple experiments.

### Pharmacological inhibitors

Cycloheximide (Sigma, St Louis, MO) was used at a final concentration of 100 µg/ml from a stock of 10 mg/ml stock solution in ethanol and added to exponential phase cultures in YE at 25°C (*Polanshek, 1977*). Latrunculin A (LatA) was used at a final concentration of 200 µM from a 100X stock in DMSO (*Chang, 1999*). LatA or cycloheximide were added to cells in a 35-mm glass bottom dish (described above) and imaged over time.

## Microscopy

Images were generally acquired using a spinning-disc confocal fluorescence NikonTI-based microscope system (Nikon Instruments, Melville, NY, Yokogawa, Tokyo, Japan, Solamere Technology, Salt Lake City, UT) with an EM CCD camera (Hamamatsu Corp, Boston, MA) and a 100X 1.4 N.A. objective with a 1.5X magnifier (*Saunders et al., 2012*). A wide-field Nikon Eclipse 800 microscope and a 60X 1.4 N.A. objective was also used for some studies. FRAP studies were performed with a Zeiss 710 scanning confocal microscope.

## Image analysis

ImageJ (NIH) and custom MatLab (Mathworks, Natick, MA) software were used for analysis.

## Pom1p gradient analyses

Fluorescence intensity values around the cortex of cells were measured from images of cells in a medial focal plane, using custom MatLab software for the automated generation of a one-pixel wide mask around the cell cortex, followed by manual correction (*Saunders et al., 2012*). Time-averaged images of pom1-fusions used average projections of 50 0.5 s frames over 25 s. The average pom1-tomato intensity at the medial cortex was measured in a 3-pixel wide by 3 μm long rectangle over the medial cortex, and the mean background value outside of the cells was subtracted. To measure the pom1p gradient decay lengths, cells expressing the appropriate pom1-fusion were imaged for 3 s in a single confocal section through the middle of the cell. Cells were segmented as described in *Saunders et al., 2012*. Intensities were normalized to one at the cell tip and background subtraction performed such that the different fusions had zero intensity 5 μm from the tip. Curves were then fitted to $\exp(-x/\lambda)$, where $\lambda$ is the decay length of the profile, with $\lambda$ shown in *Figure 1—figure supplement 4B*.

## Cdr2p node analyses

Cdr2-GFP intensity was quantified using six different methods (*Figure 2—figure supplement 3*). (A) Maximum projections were made of 13 slices of confocal sections taken 0.4 μm apart. A region of interest (ROI) was selected in ImageJ by hand around the cdr2p nodes, excluding as much background as possible. The area and total intensity of the ROI was recorded, and the ROI width was determined by the spread of cdr2p nodes along the long axis of the cell. (B) Similar to (A) but the maximum projection was taken from the top three slices consisting of the 'top' cortical section of the cell. (C) Similar to (A), except the ROI was selected by an image analysis program in Matlab (custom-written) which selected only pixels over a predetermined threshold (approximately two times the mean background intensity). In this case, the width was not determined. (D) Maximum projections were taken similar to (A). We used the Find Maxima macro function in ImageJ to find the brightest pixel from a local intensity source (likely nodes), counting their number and totaling their intensity to estimate total intensity levels. In this case, width was also not determined. (E) We used a single confocal section through the middle of the cell and acquired images over 30 s. A region was then chosen for each time-averaged data set that overlapped the nodal region (now seen as a line on the perimeter of the cell) in a single pixel wide line that was 3 μm long. The intensity was measured from that line and summed. (F) Maximum projections were made of 13 slices of confocal sections taken 0.4 μm apart. Individual cells were then taken and rotated so their long axis was horizontal. A rectangular ROI was fixed at 3 μm wide and 3.72 μm tall for all cells and placed at the center of the nodal region. The mean intensity was then recorded in this fixed area.

In all these instances, the mean background intensity from an area outside of the cells was subtracted for each pixel. These different methods all resulted in the same linear increase of cdr2-GFP intensity levels in the nodal region as a function of increasing cell length. However, due to the fact that each method measured cdr2-GFP levels in different ways, the exact slope and variance of the correlation differed from method to method.

The single cell analyses of cdr2-GFP in the wildtype (*Figure 2E,F*) used 13 confocal sections 0.4 μm apart. Intensities were measured in a hand drawn ROI that contained the majority of cdr2p nodes and the mean background outside the cells was subtracted.

In *Figures 3 and 4*, in analysis of LatA and cycloheximide-treated cells, and *for3Δ* cells, maximum projections of Z-stacks comprising 13 confocal sections 0.4 μm apart were used. Intensities were measured in a hand drawn ROI that contained the majority of cdr2p nodes and the mean cytoplasmic value inside the cells at each time point was subtracted.

In the measurements of rates of growth and cdr2-GFP accumulation in *for3* mutants (*Figure 4C,D*), growth rates were calculated by a least squares linear fit to the cell length as a function of time (images every 30 min), over 60–120 min. The rate of change in nodal cdr2p intensity with time was also calculated by a least squares linear fitting. In both the cases, the error on the fit was found for each cell. To test whether a positive correlation between growth and cdr2p accumulation rates was robust, we performed numerical simulations using the distributions of the measured rates, and their errors, to create in silico data. Corresponding to each pair of values in the measured data set, we created a new in silico pair by drawing from Gaussian distributions with widths given by the measured errors in growth rate and cdr2p accumulation rate. We then performed a linear least squares fitting on each in silico data set to find the level of correlation and test whether it was greater than zero—that is whether a positive correlation existed between cdr2p accumulation rate and cell growth rate. Repeating this process $10^6$ times, we found a probability of ~0.0005 that a positive correlation would be absent. Hence, our conclusion of a positive correlation between the cdr2p accumulation rate and cell growth rate is robust. Results shown in *Figure 4D* are for a single experiment (n = 21 cells); similar results were found in multiple additional experiments (data not shown).

Protein counts were estimated by quantitative fluorescence intensity in ratios with standard proteins that had been quantitated previously (*Wu and Pollard, 2005*; *Coffman and Wu, 2012*). GFP-MotB complexes in live bacteria were used as a standard at 22 GFP molecules/dot (*Leake et al., 2006*; *Coffman et al., 2011*; *Laporte et al., 2011*).

To calculate the width of the nodal cdr2p region, we fitted the function $\left(ae^{-(x-x_0)^2/2\sigma^2} + b\right)$ to the cdr2p profile from a time-averaged (90 s) confocal section through the middle of each cell (385 cells). We only analyzed cells with a good quality of fit (so that the measured σ is meaningful) and with σ >0.5 μm (thereby excluding cells with distorted fits due to one very bright nodal region). This process left 237 cdr2p intensity profiles for analysis. Each cell was binned according to length (8–9 μm, 9–10 μm, …) and the mean and standard deviation calculated within each bin, see *Figure 2C*.

For the cortical Cdr2-GFP profiles shown in *Figure 7C*, *Figure 7—figure supplement 1C*, a cortical mask was extracted as described in *Saunders et al. (2012)*. The center of each cell was located and the angles between a chosen tip and each pixel on the mask were calculated (so a pixel at the opposing tip would have angle π). Angles were then binned into 100 sectors from 0 to 2π and the mean cdr2-GFP intensity at a given angle around the cell was calculated. Angles were converted into the mean distance from the tip by assuming that in the mid plane the cell can be approximated as two semicircles connected by straight lines, using the mean cell length and radius for each cell type. For *pom1Δ* cells, the tip with the highest cdr2-GFP intensity was defined to be at d = 0 μm.

To calculate total cortical signal the sum of the cdr2-GFP signal on the mask was used (*Figure 7F*). For analysis of cdr2-GFP intensity in a 3-μm cortical region around the cell middle (*Figure 7G*, *Figure 7—figure supplement 1D*) each pixel in the cell cortical mask within ± 1.5 μm of the cell centre was identified and then the cdr2-GFP was summed over only these pixels.

## Measuring cell surface area and volume

For *Figure 6*, cells were grown in liquid YE5S media at 25°C, and imaged on agarose pads. Cell surface area and volumes were measured using manual segmentation. In *Figure 6B,C*, we used a single mid focal plane brightfield image, whereas in *Figure 6D,E*, we used a single mid focal plane of a fluorescent image of blankofluor-stained septated cells. Cell perimeters were manually traced, with the mean surface area, $A_{cor}$ and mean volume, V, calculated in Matlab assuming radial symmetry around the long axis of the cell (as the cross-section of fission yeast cells are nearly circular). To compare cells of similar surface area, we selected all cells with surface areas in the range $A_{cor}$ ± 0.10–0.20 $A_{cor}$. The range of ± 10–20% was taken to ensure we had enough cells included for statistical significance (between 24 and 32 cells), but that the range was reasonably constrained. The specific range was adjusted for each subset of cells for the different cell lines such that the mean surface area (*Figure 6B*) or mean volume (*Figure 6C*) were equal to within ± 1%. The unbinned data for each cell type is shown in *Figure 6—figure supplement 1*. Likewise, for comparing cells of similar volume, we included all cells with volume in the range V ± 0.10–0.20 V. For measuring cell size at septation (*Figure 6D,E*), cells without cdr2-GFP were analyzed. The septum was not included in this analysis, as we wanted to extract cellular dimensions at entry into mitosis prior to septum formation. We also analyzed a separate data set with the cdr2-GFP strains (n > 45 cells for each genotype), and a data set using brightfield images, which all showed the same behavior.

The similarity of the distributions for cell length, surface area and volume at mitosis were compared in the wild-type, *rga2Δ* and *rga4Δ* mutants using the Jensen–Shannon distance (*Figure 6E*). The Jensen–Shannon distance is a statistical measure that quantitatively compares the overlap of two or more distributions, with a distance of 1 corresponding to the distributions having no shared information and a distance of 0 to identical distributions. The Jensen–Shannon distance is the square root of the Jensen–Shannon divergence, which is defined in terms of the Shannon entropy function of the probability distributions (see *Lin, 1991*).

## Description of mathematical modeling

### Cell morphology

Wild-type fission yeast geometry is approximated as a cylindrical body with hemispherical caps at either end. The radius of the cell is approximately constant at about $R = 1.5$ μm, while over the cell cycle the cell length grows from about $L = 7$ μm to $L = 14$ μm. In this case, the surface area $A_{cor}$ and length $L$ are strictly proportional and related by $A_{cor} = 2\pi RL$. This approximation of the cell morphology predicted surface areas and volumes consistent with experimentally measured values (data not shown).

### Timescales

From cdr2p FRAP experiments, the lifetime of the nodal cdr2p is on the order of 3 min (*Figure 2—figure supplement 4*). From live imaging of cortical cdr2p (*Figure 5A*), cortical cdr2p dynamics are rapid, with a timescale on the order of seconds. However, cell growth is considerably slower (with doubling times on the order of hours) and hence we solve the subsequent equations for cdr2p with each cell size considered to be in quasi-steady-state.

## Model I

### Different forms of cdr2p

In our first model cdr2p is taken to have three forms: cytoplasmic, cortical, and nodal. (1) Cytoplasmic cdr2p has a homogeneous concentration, $\rho_{cyt} = N_{cyt}/V$, which does not change significantly with cell size, as found experimentally (*Figure 5—figure supplement 1A*). (2) Cytoplasmic cdr2p can associate with the membrane. For this cortical cdr2p population, $N_{cor}$ denotes the copy number and $\rho_{cor} = N_{cor}/A_{cor}$ is the corresponding concentration. (3) The cortical cdr2p can cluster in nodes at the midcell cortex. For this nodal cdr2p population, $N_{nod}$ denotes the copy number, with corresponding concentration $\rho_{nod} = N_{nod}/A_{nod}$, where $A_{nod}$ is the area of the cell membrane occupied by the nodes.

We employ two approaches in our analysis. First, we take the cortical cdr2p population as diffusing rapidly and hence having an approximately uniform distribution around the cell membrane. The nodal cdr2p is taken to be uniformly distributed within the nodal region, though below we also consider a model variant where we explicitly consider diffusion of the cortical cdr2p population.

### Uniform cdr2p populations

Here, the uniformly distributed cdr2p populations in quasi-steady-state are described by the following equations:

$$0 = \beta \frac{A_{cor}}{V} N_{cyt} - \nu N_{cor} - \alpha \frac{A_{nod}}{A_{cor}} N_{cor}$$

$$0 = \alpha \frac{A_{nod}}{A_{cor}} N_{cor} - \eta N_{nod},$$

where, $\beta$ is the association parameter of cytoplasmic to cortical cdr2p, $\nu$ is the disassociation rate of cortical cdr2p back into the cytoplasm, $\alpha$ is the rate of uptake of cortical to nodal cdr2p and $\eta$ is the disassociation rate of nodal cdr2p back into the cytoplasm. These equations can be solved exactly:

$$\rho_{nod} = \rho_{cyt} \frac{\beta}{\eta} \left( \frac{\nu}{\alpha} + \frac{A_{nod}}{A_{cor}} \right)^{-1} \quad \text{and} \quad \rho_{cor} = \rho_{nod} \frac{\eta}{\alpha}.$$

The value of $\beta$ is not important as it only enters our solutions as a constant prefactor. The rate of cdr2p disassociation from the nodes back into the cytoplasm, η, can be estimated from our FRAP

experiments (*Figure 2—figure supplement 4*). We find that the cdr2p has a nodal occupancy time of around 3 min. We can therefore estimate $\eta = 5 \times 10^{-3} \text{s}^{-1}$.

Much higher concentrations of cdr2p in the nodes are experimentally observed than elsewhere on the cortex. From above, since $\rho_{nod}/\rho_{cor} = \alpha/\eta \gg 1$, we therefore require that the rate $\alpha$ of uptake of cortical cdr2p into the nodes be considerably greater than the rate $\eta$ of nodal cdr2p disassociation back into the cytoplasm. This constraint places a lower bound on $\alpha$, and consistently we choose $\alpha = 1.0 \text{ s}^{-1}$.

Experimentally, we observe significant scaling of $\rho_{nod}$ with increasing $A_{cor}$ (or equivalently with cell length in the wild type). For this to occur, our model requires that two key criteria be met. First, the cortical cdr2p must be much more likely to be taken up into the nodes than disassociate from the cortex, that is from above $v/\alpha \ll 1$. Since $\alpha$ is already constrained, we have a further restriction on $v$. Accordingly, we choose $v = 5 \times 10^{-3} \text{s}^{-1}$, meaning that the cdr2p disassociation rates from the nodes and cortex are the same. Second, $A_{nod}$, the area of the nodal region, must not scale proportionally with $A_{cor}$, the total cell area, as the cell size increases. Importantly, this model requirement was verified experimentally, see *Figure 2C*. Although the cdr2p does spread to an extent during growth (*Figure 2C*), the majority of nodal cdr2p is localized to the center of the cortex (*Figure 1—figure supplement 1A*). Therefore, we take $A_{nod}$ to be constant in the *Figure 5E* fitting. In *Figure 5—figure supplement 1*, we also include the effect of $A_{nod}$ increasing with cell length (see below).

Finally, the model also included the experimentally observed (slight) decrease in $\rho_{cyt}$ as a function of cell length, *Figure 5—figure supplement 1A* black line (gradient = −0.01 μm⁻¹ after normalization to the average cytoplasmic intensity). However, there was little difference in our results between this case and when assuming a strictly constant $\rho_{cyt}$ (data not shown).

As shown in *Figure 5E*, the above model can recapitulate the observed cdr2p scaling. The larger the value of $\alpha$, the stronger the scaling effect will be, as more cortical cdr2p—which effectively 'measures' the cell area—is taken up into the nodes—which effectively 'read-out' the area measurement.

## Cdr2p membrane localization does not necessarily imply cell size control

If cytoplasmic cdr2p can only associate to the membrane by being directly taken up by nodes at mid-cell then, we can simply leave out the cortical cdr2p form. By balancing the cdr2p coming onto the nodes (parameter $\beta$) with that disassociating (rate $\eta$) we find $\rho_{nod} = (\beta/\eta) \rho_{cyt}$. The concentration of nodal cdr2p is now independent of cell size, assuming $\rho_{cyt}$ is constant. A similar conclusion is reached if the nodes can form anywhere on the cortex by direct association of cdr2p from the cytoplasm. In both cases, the system cannot sense cell area because both association and disassociation occur over the same region. To sense cell area we require one process, which here is the association of cdr2p anywhere onto the membrane, to scale proportionally with cell area. However, the second process, which here is the uptake of cortical cdr2p into the nodes, must be localized over a region whose size does not scale proportionally with the total cell area as the cell grows. The outcome is then a density (for both $\rho_{cor}$ and $\rho_{nod}$) that scales with total cell membrane area.

## Incorporating cortical cdr2p diffusion

Incorporating cortical diffusion into the model is straightforward, though solutions now need to be found numerically. We assume that the cytoplasmic cdr2p is still homogeneous and at an almost constant concentration, decreasing only slightly with cell length as described above. Assuming that on the membrane the cdr2p densities only depend on the long-axis coordinate, $x$, the equations become, at quasi-steady-state:

$$0 = D_{cor} \frac{\partial^2 \rho_{cor}}{\partial x^2} - v\rho_{cor} - \alpha(x)\rho_{cor} + \beta\rho_{cyt}$$

$$0 = D_{nod} \frac{\partial^2 \rho_{nod}}{\partial x^2} + \alpha(x)\rho_{cor} - \eta\rho_{nod},$$

with $\beta$, $\eta$ and $v$ taking the same values as before (*Figure 5D*). We now incorporate the width of the nodal region by using an association function $\alpha(x)$ for the uptake rate of cortical to nodal cdr2p. This scheme is, of course, a simplification of the true uptake dynamics, which presumably involve cdr2p aggregation and clustering. Nevertheless, this simplification is sufficient for understanding the mechanistic basis of size scaling. We take $\alpha(x) = \alpha_0 \exp(-x^2/2\omega^2)$, with $\alpha_0 = 0.5 \text{ s}^{-1}$. Here, $\omega$ is the fitted width

of the nodal region (fitted to the appropriate data set in *Figure 2C*): $\omega = a\,(1-e^{-L/s})$ where s = 7 μm and a = 2.2 μm. In *Figure 5A*, we see that the nodal cdr2p does not move significantly over an extended period, suggesting $D_{nod}/D_{cor} \ll 1$. Therefore, we set $D_{nod} = 0$.

For the nodal cdr2p density to serve as a read-out of the entire cell membrane area, the typical cortical cdr2p diffusional displacement along the long cell axis must be greater than 5 μm. This requirement ensures that cortical cdr2p can diffuse along the long axis from the cell tips to the nodal region without first disassociating. Hence $\sqrt{(2D_{cor}\,\tau)} > 5$ μm, where $\tau$ is the lifetime of cortical cdr2p. Given a cortical cdr2p lifetime of around 3 min (see above), this implies that the diffusion constant should be greater than about $D_{cor} = 0.1$ μm²s⁻¹. In our simulations, we use $D_{cor} = 0.2$ μm²s⁻¹, but if the lifetime of the cortical cdr2p is shorter, then the cortical diffusion constant will need to be larger.

We solve the equations numerically in one-dimension with length L and hard wall boundary conditions, using Matlab. This model can reproduce the profile of cortical/nodal cdr2p for different cell lengths (or equivalently with cell surface areas in the wildtype) (*Figure 5—figure supplement 1B-D*). The increase in nodal cdr2p concentration as the cell grows (*Figure 2F*) is also captured (*Figure 5—figure supplement 1E,F*). In conclusion, within reasonable parameter ranges, the model prediction—that the concentration of cdr2p in the nodes increases with cell area—is robust to the inclusion of cortical cdr2p diffusion.

## Model II

A simple alternative model for area scaling involves cdr2p becoming modified (e.g., phosphorylated). We assume that unmodified cdr2p diffuses rapidly in the cytoplasm, with homogeneous density $\rho_{cyt} = N_{cyt}/V$. Unmodified cdr2p can then bind to the membrane with a binding constant β. Once present on the membrane, with a correspondingly homogeneous density $\rho_{cor} = N_{cor}/A_{cor}$, cdr2p can unbind back into the cytoplasm at a rate ν, while at the same time becoming modified (e.g., phosphorylated). This cytoplasmic, modified form of cdr2p, with homogeneous density $\rho^*_{cyt} = N^*_{cyt}/V$ can rapidly diffuse, and then bind to the nodal region on the cortex, with a binding constant α, or spontaneously become unmodified at a rate μ. Finally, modified cdr2p in the nodal region, with density $\rho_{nod} = N_{nod}/A_{nod}$, can unbind and become unmodified cytoplasmic cdr2p at a rate η (*Figure 5*, *Figure 5—figure supplement 2A*). The corresponding steady-state equations are:

$$0 = \beta \frac{A_{cor}}{V} N_{cyt} - \nu N_{cor}$$

$$0 = \nu N_{cor} - \mu N^*_{cyt} - \alpha \frac{A_{nod}}{V} N^*_{cyt}$$

$$0 = \alpha \frac{A_{nod}}{V} N^*_{cyt} - \eta N_{nod}.$$

In the case without spontaneous reversion of cytoplasmic, modified cdr2p back to its unmodified form (i.e., μ = 0), we can solve these equations to find

$$\rho_{nod} = \left(\frac{\beta}{\eta}\right)\left(\frac{A_{cor}}{A_{nod}}\right)\rho_{cyt}.$$

Experimentally, we observe significant scaling of $\rho_{nod}$ with increasing $A_{cor}$ (or equivalently with cell length in the wildtype). This is in agreement with Model II, again provided that $A_{nod}$, the area of the nodal region, does not scale proportionally with $A_{cor}$, the total cell membrane area, as the cell size increases. If there is spontaneous reversion of cytoplasmic, modified cdr2p back to its unmodified form, then the above solution for $\rho_{nod}$ becomes

$$\rho_{nod} = \left(\frac{\beta}{\eta}\right)\left(\frac{A_{cor}}{A_{nod}}\right)\left(\frac{1}{\frac{\mu V}{\alpha A_{nod}} + 1}\right)\rho_{cyt}.$$

Provided $\alpha A_{nod} \gg \mu V$, then the reversion process can be neglected and our cell area scaling results are unchanged. As in Model I, the larger the value of α, the stronger the scaling effect will be, as more

modified, cytoplasmic cdr2p—which effectively 'measures' the cell area—is taken up into the nodes—which effectively 'read-out' the area measurement.

As above with Model I, the value of $\beta$ is not important as it only enters our solutions as a constant prefactor. The value of $v$ is also not important for the behavior of $\rho_{nod}$ (see above), though we use a value of $v = 0.5$ s$^{-1}$ to ensure low levels of cortical cdr2p, as observed experimentally. We again use the FRAP data to estimate $\eta = 5 \times 10^{-3}$s$^{-1}$. Since $\rho_{nod} = (\alpha/\eta) \, \rho^*_{cyt}$, we take $\alpha = 0.5$ $\mu$ms$^{-1}$ so that the concentration of modified cdr2p in the nodal region is relatively large compared to that in the cytoplasm (required since there is no observed scaling of cytoplasmic cdr2p concentration with cell length). Further, we use a low rate of spontaneous modification loss $\mu = 0.03$ s$^{-1}$, so that the lifetime of modified, cytoplasmic cdr2p is relatively long. The fitting of this model to the experimental scaling of cdr2p is shown in (*Figure 5*, *Figure 5—figure supplement 2B*).

In this model, the mechanism of size scaling is similar to that of model I. One process, the unbinding of cortical cdr2p can occur from anywhere on the membrane, and so scales proportionally with cell area. However, a second process, in this case the uptake of modified, cytoplasmic cdr2p into the nodes, occurs over a region whose area does not scale proportionally with the total cell area as the cell grows. The outcome is again a density (for both $\rho^*_{cyt}$ and $\rho_{nod}$) that scales with total cell membrane area. The presence of the modified form of cdr2p is vital, so that information about membrane area can be protected and relayed to the nodes without being lost into the general cytoplasmic cdr2p population, which does not show size scaling characteristics (*Figure 5—figure supplement 1A*).

## Role of pom1p in nodal cdr2p scaling

The model does not explicitly include pom1p. However, this does not mean that pom1p is unimportant in the regulation of nodal cdr2p. As the *pom1Δ* experiments demonstrate, without pom1p acting as a tip inhibitor for cdr2p, nodal cdr2p can form in an extended part of the cell, with such cells observed to divide at shorter lengths. Conversely, in pom1p mutants where pom1p is mistargeted all over the plasma membrane, cdr2p does not form localized regions of high nodal concentration, with such cells dividing at longer lengths. Therefore, pom1p appears to play an important role in defining the region of nodal cdr2p accumulation, without which the cdr2p-dependent control of cell length is perturbed. Accordingly, pom1p is implicitly included in the model by defining a spatially limited region that can be occupied by the cdr2p nodes.

## Acknowledgements

We thank Chang and Howard group members, Lars Hufnagel and his group for support and discussion, Neal Padte for preliminary data, the Columbia University Department of Microbiology and Immunology scanning confocal facility, Zhou Zhou for image analysis, Kathy Gould, James Moseley, Rafael Daga, and Jian Qiu Wu for strains, and Anna Kazatskaya and Orna Cohen-Fix for discussion.

## Additional information

### Funding

| Funder | Grant reference number | Author |
| --- | --- | --- |
| National Institutes of Health | GM056836 | Fred Chang |
| Biotechnology and Biological Sciences Research Council | BB/J004588/1 | Martin Howard |
| EMBL Interdisciplinary Postdoc Programme | | Timothy E Saunders |
| Singapore National Research Foundation | | Timothy E Saunders |
| MEC/ Fulbright Postdoctoral Award | | Ignacio Flor-Parra |

The funders had no role in study design, data collection and interpretation, or the decision to submit the work for publication.

## Author contributions

KZP, TES, MH, Conception and design, Acquisition of data, Analysis and interpretation of data, Drafting or revising the article; IF-P, Acquisition of data, Analysis and interpretation of data; FC, Conception and design, Analysis and interpretation of data, Drafting or revising the article

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
