## [Decision Letter]

[Editors’ note: a previous version of this study was rejected after peer review, but the authors submitted for reconsideration. The two decision letters after peer review are shown below.]

Thank you for choosing to send your work entitled “Cortical regulation of cell size by a sizer cdr2p” for consideration at *eLife*. Your full submission has been evaluated by a Senior editor and 3 peer reviewers, one of whom is a member of our Board of Reviewing Editors, and the decision was reached after discussions between the reviewers (none of whom are connected with the previously published papers on this subject). We regret to inform you that your work will not be considered further for publication at this time.

This is primarily due to the fact that the revisions that need to be done and the advance required are so extensive that we believe they cannot be accomplished within a reasonable period of time.

As you will see the 3 referees feel that your proposal of an alternative mechanism for size control is very interesting and are keen to see this developed further.

Although referee #3 (an expert in high resolution microscopy and image analysis) is the most positive of the three, in further consultations between the 3 expert referees this referee agreed that the insight on the new mechanism provided by your work was not fully developed. This referee has pointed out ways in which image analysis can be improved.

Below are a number of points the referees have raised (reproduced verbatim), addressing which will make the paper potentially suitable for publication in *eLife*.

*Reviewer*
*#1:*

The manuscript by Pan et al. addresses the interesting and important question of how cell size homeostasis is established using fission yeast as a model, since fission yeast cells divide uniformly at a fixed size. The size at division is a manifestation of the size at which cells enter M phase. Influential previous work from Martin and Nurse laboratories have argued that a gradient of the protein kianse Pom1 serves to inhibit the medially localized Cdr2 until the Pom1 concentration reaches a low level in the Cdr2 containing zone and causing Cdr2 activation. This in turn fixes cell size at division and links growth to division.

The main claim of the Chang paper is that Pom1 cannot function in a gradient, since using improved image analysis tools such gradients are not seen in the current study. Pan et al. attribute the gradients observed by Nurse and Martin groups as a consequence of Z-projection of the entire cell fluorescence into a single line, whereas Pan et al. are only looking at cortical Cdr2.

I am not entirely sold on the Nurse/Martin models and I believe thus an alternative mechanism for cell size control is certainly welcome. That said, the authors do not advance their model (i.e., the number of Cdr2 nodes themselves function as a ruler) sufficiently to merit publication in *eLife*.

Some of the main issues:

1) It will be important for the authors to provide a mechanism of how increased Cdr2 node numbers contributes to mitotic entry. One scenario that can be imagined is that Cdr2 is constitutively active and its concentration in the cell middle regulates mitotic entry. Is this the case? Other scenarios can be imagined as well.

2) The overexpression of *cdr2* and its shut down should be done with GFP tagged versions to see if the model Pan et al. are advancing holds up. i.e the reduced size at division is a direct effect of increased Cdr2 nodes and vice versa (as opposed to protein levels directly affecting division length).

3) The Nurse and Martin studies have performed a vast array of genetic epistasis experiments. How is the model proposed by Pan et al. reconciled with the published epistasis data.

4) It will be important for Pan et al. to express either the PMT1-Pom1 fusion (Nurse paper) or Tea1-Pom1 fusion (Martin paper) and see whether node numbers are reduced the way Pan et al. carry out image processing. This was a key experiment in the Nurse and Martin lab papers.

5) The *for3* experiment is interesting, but cell size at birth has not been factored into the experiment, which can complicate interpretation.

6) The relationship between *cdr2* and *pom1* needs to be established better. For example is there a precocious accumulation of Cdr2 nodes (albeit at wrong locations) at premature mitotic entry. If so, this says that the Pan et al. mechanism needs to be modified since the proximity of Cdr2 nodes to the cell middle does not seem to influence timing of mitotic entry.

In summary, an interesting alternative model that needs to be developed a lot further for publication in *eLife*.

*Reviewer*
*#2:*

This manuscript uses the fission yeast *S. pombe* to investigate how cells divide at a precise size. Two previous papers from the Moseley/Nurse and Martin labs proposed a model for how *S. pombe* measures size. The model is based on the observed spatial separation of two antagonistic regulators of cell size (Cdr2 and Pom1). Cdr2 promotes mitotic entry and localizes to medial cortical nodes and Pom1 antagonizes Cdr2 and localizes in a gradient emanating from cell tips. The model proposes that increase in cell length causes reduction in medial Pom1 and activation of Cdr2/mitotic entry. This manuscript challenges the existing model by showing that the concentration of Pom1 at the medial cortex does not change as cells elongate. Based on their data showing that the concentration of Cdr2 at nodes increases with cell size (even though the overall concentration of the protein does not change significantly) the authors propose that cells “measure” size through levels of medial Cdr2. A mathematical model is described to explain the size dependent increase in Cdr2 node concentration. Central to this model is the assumption that Cdr2 must first load onto the membrane from where it can laterally diffuse to medial nodes and be incorporated into node structures. In this way, the concentration of Cdr2 in medial nodes would depend on the surface area of the cell.

The strength of the paper is its data challenging the central premise of the existing model, and the discovery of length dependent increase in Cdr2 levels. The major weakness of the paper is the lack of data supporting their alternative model. I am sure that there are many other alternative models that (in absence of supporting data) could be mathematically modeled. Thus I think that for the manuscript to constitute a major advance, there needs to be some test of the model. Listed below are several questions I have about the model and potential ways to test it.

Specific points:

1) The assumption that Cdr2 must bind membrane and then diffuse “rapidly” to medial nodes does not make sense to me. Why couldn't Cdr2 go directly from the cytoplasm to nodes, since diffusion is much faster in the cytoplasm than in the plasma membrane.

2) Comparison of Cdr2 node levels in round cells to that in cylindrical cells of the same volume should be informative. Because the surface area is greater in cylindrical cells, the levels of Cdr2 in nodes should be higher in the cylindrical cells.

3) Although Cdr2 levels in medial nodes increase with cell size, the increase is quite small, and I wonder whether such a small increase is sufficient to trigger mitotic entry.

4) Because nodal Cdr2 spreads away from the cell middle in *pom1Δ* mutants, the proposed model would predict that these cells should enter mitosis at a larger size because Cdr2 is farther away from its target Wee1 at the SPB. In fact *pom1Δ* cells divide at reduced cell size. Some effort to incorporate Pom1 into the model seems warranted.

*Reviewer*
*#3:*

This paper by Pan et al. describes a new model for cell-size control in fission yeast. Cell-size control is a fundamental and difficult question for all cell types. The fission yeast *S. pombe* has a regular rod shape and divides at a reproducible cell length and a constant cell diameter. Combined with other advantages as a model system, the fission yeast is one of the best systems to study the molecular mechanisms of cell-size control. Previous studies have proposed an elegant and simple model for the regulation of cell size: Pom1 polar gradient and Cdr2 kinase-based interphase nodes work together to control mitotic entry and cell size. One line of key evidence of the model is that the Pom1 polar gradient has highest concentration at the cell pole and lowest concentration in cell middle, which decreases as cells grow. When cells reach a certain length, Pom1 gradient cannot inhibit Cdr2 kinase activity in the nodes, then Cdk1 kinase becomes active and cells enter mitosis.

The new work from Pan et al. proposes a new Cdr2-based model based on rigorous data analyses and mathematical modeling. They find that Pom1 concentration at cell middle does not change during cell growth, instead the Cdr2 concentration increases with cell length. They pointed out that nuclear exclusion of Pom1 accounts for the differences. The data in this high quality work are very convincing and well supported by the careful controls. I do not have major concerns for this elegant study.

The minor concerns listed below are not essential for the main conclusion of this paper.

1) I do not understand why the mean background outside the cells was subtracted when measuring fluorescence intensity. When measuring cortical Pom1 and Cdr2 intensity, most background in the ROI should come from diffused molecules within the cells. Thus, it might be better to use the cytoplasm as background. This will not affect the main conclusion of this study. Instead, it will make the increase of Cdr2 intensity at cell middle more obvious since Cdr2 global concentration decreases slightly when cells grow longer (Figure 4—figure supplement 1).

2) It is impressive that six methods were used to measure Cdr2 fluorescence in cortical nodes. However, five of them are based on maximum-intensity projections. The sum-intensity projections are more suitable since the total proteins in each nodes instead of maximum pixel intensity might be most important for cell-size control. This may explain the steeper slopes of all other panel except E in Figure 2—figure supplement 3.

3) In Figure 2, each cell has only about 30 to 40 nodes. This might be an underestimation due to large z spacing (∼0.4 μm). To count node number accurately, much smaller slice spacing, like 0.05 or 0.1 μm, is needed.

[Editors’ note: what now follows is the decision letter after the authors submitted for further consideration.]

Thank you for sending your work entitled “Cortical regulation of cell size by a sizer cdr2p” for consideration at *eLife*. Your article has been favorably evaluated by a Senior editor and 3 reviewers, one of whom is a member of our Board of Reviewing Editors.

The Reviewing editor and reviewers discussed their comments before we reached this decision, and the Reviewing editor has assembled the following comments to help you prepare a revised submission. Based on these discussions, we would like you to address the two points raised below.

1) One of the reviewers (#2) has made a point about computing Cdr2 gradients near the SPB and it seems reasonable to attempt this given that you have the relevant information on Cdr2 concentration. The referee's point is provided verbatim below:

“Whether increased medial Cdr2 is in fact what triggers mitotic entry is still not clear. The authors suggest that Cdr2 localization at the medial cortex could generate a concentration gradient that extends to the SPB where the Cdr2 target Wee1 resides. As Cdr2 medial concentration increases with cell elongation, it is proposed that the Cdr2 concentration at the SPB would increase until it is sufficient to inactivate Wee1. I have some concern about this idea because cytoplasmic gradients are likely to be very steep given the high rates of diffusion. Given that the authors know both the Cdr2 concentration at the medial cortex and the turnover rate they could calculate how steep the gradient from the cortex is likely to be and whether the concentration of Cdr2 around the SPB would actually change significantly as cells elongate. This would at least say whether this notion made sense. If so, it would be interesting (in future studies) to examine mutants that caused the SPB to be positioned further away from the cell cortex to see if this changed the timing of mitotic entry.

It is not clear from the main text if the model explains the existence of medial nodes, how they are distributed, and how new ones form with cell length”

2) Please also address the following point from referee #3:

“One minor improvement required for new figures (and some old figures) is that most Y-axes in the graphs should start from zero.”

With these two points addressed your paper will make a fine contribution to our understanding of cell size control and mitosis.

---

## [Author Response]

[Editors’ note: the author responses to the first round of peer review follow.]

*Below are a number of points the referees have raised (reproduced verbatim), addressing which will make the paper potentially suitable for publication in* eLife*.*

Reviewer #1:

*The manuscript by Pan et al. addresses the interesting and important question of how cell size homeostasis is established using fission yeast as a model, since fission yeast cells divide uniformly at a fixed size. The size at division is a manifestation of the size at which cells enter M phase. Influential previous work from Martin and Nurse laboratories have argued that a gradient of the protein kianse Pom1 serves to inhibit the medially localized Cdr2 until the Pom1 concentration reaches a low level in the Cdr2 containing zone and causing Cdr2 activation. This in turn fixes cell size at division and links growth to division*.

*The main claim of the Chang paper is that Pom1 cannot function in a gradient, since using improved image analysis tools such gradients are not seen in the current study. Pan et al. attribute the gradients observed by Nurse and Martin groups as a consequence of Z-projection of the entire cell fluorescence into a single line, whereas Pan et al. are only looking at cortical Cdr2*.

*I am not entirely sold on the Nurse/Martin models and I believe thus an alternative mechanism for cell size control is certainly welcome. That said, the authors do not advance their model (i.e., the number of Cdr2 nodes themselves function as a ruler) sufficiently to merit publication in* eLife*.*

*Some of the*
*main issues:*

*1) It will be important for the authors to provide a mechanism of how increased Cdr2 node numbers contributes to mitotic entry. One scenario that can be imagined is that Cdr2 is constitutively active and its concentration in the cell middle regulates mitotic entry. Is this the case? Other scenarios can be imagined as well*.

The events downstream of Cdr2 are conceptually a different problem, and are beyond the scope of this work. Our study aims to understand how the concentration of Cdr2 at the medial cortex is able to act as a cell size sensor. Given our data ruling out Pom1-based mechanisms, resolving this issue is, we believe, by far the most pressing question. How this information is subsequently read out is a separate issue. We also note that this read out was not addressed by the earlier Martin/Nurse studies. In the Discussion, we simply point out that some of the downstream targets such as Wee1 and Cdc2 are located at the spindle pole body, so that communication between Cdr2 and these targets, perhaps by concentration gradients, may be occurring. Providing additional understanding of this step, through study of Cdr2 biochemical activity, would constitute a different study.

*2) The overexpression of* cdr2 *and its shut down should be done with GFP tagged versions to see if the model Pan et al. are advancing holds up. i.e the reduced size at division is a direct effect of increased Cdr2 nodes and vice versa (as opposed to protein levels directly affecting division length)*.

Our experiments were performed with untagged Cdr2 in which mRNA measurements showed a 1.6 fold increase. We agree that it would be good to visualize Cdr2 in these cells. However, no Cdr2 antibody is available. We did try similar experiments with a Cdr2-GFP construct. This construct did not, however, show consistent cell shortening. We are concerned that this construct produces dominant negative effects when overexpressed, which cause some cell lengthening effects on top of the predicted cell shortening effects.

*3) The Nurse and Martin studies have performed a vast array of genetic epistasis experiments. How is the model proposed by Pan et al. reconciled with the published epistasis data*.

Our model is consistent with all the genetic epistasis data. We do not dispute that Pom1 is an inhibitor of Cdr2. We have included new data on quantitating Cdr2 in pom1Δ and PMT-pom1C mutants (new Figure 7).

*4) It will be important for Pan et al. to express either the PMT1-Pom1 fusion (Nurse paper) or Tea1-Pom1 fusion (Martin paper) and see whether node numbers are reduced the way Pan et al. carry out image processing. This was a key experiment in the Nurse and Martin lab papers*.

We have performed this experiment with PMT-Pom1C fusion from the Moseley paper. We find that the cortical Cdr2 intensity levels are reduced in these strains, and that these cells divide at long lengths. These data provide support for our model that the levels of Cdr2 at the medial nodes are important for its ability to regulate cell size.

*5) The* for3 *experiment is interesting, but cell size at birth has not been factored into the experiment, which can complicate interpretation*.

The cell sizes are initially very similar, as these cells divide largely medially. Moreover, we show time lapse images of some representative cells in Figure 4. Growth rates are generally not dependent on cell size in fission yeast (as shown by near-linear growth curves). Overall we do not believe the cell size at birth issue is a concern for our *for3* experiment analysis.

*6) The relationship between* cdr2 *and* pom1 *needs to be established better. For example is there a precocious accumulation of Cdr2 nodes (albeit at wrong locations) at premature mitotic entry. If so, this says that the Pan et al. mechanism needs to be modified since the proximity of Cdr2 nodes to the cell middle does not seem to influence timing of mitotic entry*.

To address this point, we have added new data that quantitate Cdr2 in *pom1Δ* mutants and in a mutant where Pom1 is mis-targeted. In the *pom1Δ* cells, there are more cortical Cdr2 nodes all around half the cortex. The levels of Cdr2 in the medial region are actually decreased. Thus, the small size of *pom1Δ* cells suggests that Cdr2 may signal, at least to an extent, from all over the cortex. This is not inconsistent with the notion that Cdr2 coming off from the nodes is activating the downstream steps on the nucleus in this

*In summary, an interesting alternative model that needs to be developed a lot further for publication in* eLife*.*

We have also added important new data showing that Cdr2 and cell size at mitosis scale with cell surface area (Figure 6), rather than, for example, volume. This was done in part in response to Reviewer 2 (see responses to the Editor and Reviewer 2). This data not only significantly supports our modeling, it also strengthens the broader concept that cells may sense their surface area. We feel that we have now significantly improved the manuscript and believe that it is now suitable for *eLife*.

Reviewer #2:

*This manuscript uses the fission yeast* S. pombe *to investigate how cells divide at a precise size. Two previous papers from the Moseley/Nurse and Martin labs proposed a model for how* S. pombe *measures size. The model is based on the observed spatial separation of two antagonistic regulators of cell size (Cdr2 and Pom1). Cdr2 promotes mitotic entry and localizes to medial cortical nodes and Pom1 antagonizes Cdr2 and localizes in a gradient emanating from cell tips. The model proposes that increase in cell length causes reduction in medial Pom1 and activation of Cdr2/mitotic entry. This manuscript challenges the existing model by showing that the concentration of Pom1 at the medial cortex does not change as cells elongate. Based on their data showing that the concentration of Cdr2 at nodes increases with cell size (even though the overall concentration of the protein does not change significantly) the authors propose that cells “measure” size through levels of medial Cdr2. A mathematical model is described to explain the size dependent increase in Cdr2 node concentration. Central to this model is the assumption that Cdr2 must first load onto the membrane from where it can laterally diffuse to medial nodes and be incorporated into node structures. In this way, the concentration of Cdr2 in medial nodes would depend on the surface area of the cell*.

*The strength of the paper is its data challenging the central premise of the existing model, and the discovery of length dependent increase in Cdr2 levels. The major weakness of the paper is the lack of data supporting their alternative model. I am sure that there are many other alternative models that (in absence of supporting data) could be mathematically modeled. Thus I think that for the manuscript to constitute a major advance, there needs to be some test of the model. Listed below are several questions I have about the model and potential ways to test it*.

We have added important new data that tests our modeling, as requested by the reviewer. The most important test is that by measuring cells with different shapes, we show that Cdr2 scales with surface area and not volume, and that cell size at division also correlates with surface area (Figure 6). This is an interesting and highly significant validation of a key model prediction. We also note that even if details of the models are not completely correct, our result that surface area may be being sensed will have broad impact. We thank this reviewer for encouraging this very productive and interesting line of experiments.

We agree that there are always ever more complex models that can be imagined to work. The aim of this work was to propose and then test simple models that have provided us with a conceptual breakthrough of how size sensing could function. It has not been at all intuitive how such size sensing might work, and we were delighted that our simple models have proven sufficient to explain the data.

*Specific*
*points:*

*1) The assumption that Cdr2 must bind membrane and then diffuse “rapidly” to medial nodes does not make sense to me. Why couldn't Cdr2 go directly from the cytoplasm to nodes, since diffusion is much faster in the cytoplasm than in the plasma membrane*.

A simple model in which Cdr2 only diffuses from the cytoplasm directly to the nodes would not result in scaling (described in the modeling section in the Materials and methods). Rather, to act as a size sensor, Cdr2 needs to bind to the membrane. In principle, subsequent transport of Cdr2 to the nodes could be either along the membrane (as in the original model) or through the cytoplasm. Accordingly, in this revision, we have added a slightly more complicated alternative model in which Cdr2 is modified on the membrane, and then may diffuse through the cytoplasm to arrive at the nodes. This model also provides scaling, and illustrates another way to think about how surface area may be sensed. However, we emphasize that the underlying mechanisms of both models are very similar, a point we make both in the main manuscript and the model discussion in the Materials and methods.

*2) Comparison of Cdr2 node levels in round cells to that in cylindrical cells of the same volume should be informative. Because the surface area is greater in cylindrical cells, the levels of Cdr2 in nodes should be higher in the cylindrical cells*.

We thank the reviewer for this excellent suggestion. We have done this experiment with “fat” and “thin” mutants. These new data, which are included in Figure 6, show that Cdr2 scales with surface area and not volume. We also added a similar experiment to show that cell size at division may also be dictated by surface area and not volume or length. Together, these data give important support for our model prediction that cells are sensing surface area, and may be doing so using Cdr2.

*3) Although Cdr2 levels in medial nodes increase with cell size, the increase is quite small, and I wonder whether such a small increase is sufficient to trigger mitotic entry*.

We do not agree that the increase is small: it is up to a factor of two, which is what would be expected for a mechanism that is essentially probing membrane area. We do, however, acknowledge that there are likely to be other factors contributing to mitotic entry that may help to form a switch-like behavior.

*4) Because nodal Cdr2 spreads away from the cell middle in pom1Δ mutants, the proposed model would predict that these cells should enter mitosis at a larger size because Cdr2 is farther away from its target Wee1 at the SPB. In fact pom1Δ cells divide at reduced cell size. Some effort to incorporate Pom1 into the model seems warranted*.

We have measured Cdr2 levels in *pom1* mutant cells and in mutants in which pom1 is spread throughout the cortex (now presented in a new Figure 7). In pom1Δ cells, we see increased levels of cortical Cdr2 all over the cortex away from medial regions. One simple interpretation of our data is that cortical cdr2p nodes all over the cell can to an extent regulate the cell cycle. Another factor, though, may be that pom1 also regulates some aspect of cdr2p activity by direct phosphorylation. We have also added an analysis of Cdr2 in PMC-Pom1C expressing cells, which exhibit low levels of cdr2p all over the cortex (Figure 7—figure supplement 1).

We did not include pom1p explicitly in the modeling because our understanding of pom1p is not complete, and, more importantly, because it is not explicitly needed to explain nodal cdr2 scaling. However, we have now included a discussion of the role of pom1 in the model within the model description in the Materials and Methods section to specifically address this point.

Reviewer #3:

*This paper by Pan et al. describes a new model for cell-size control in fission yeast. Cell-size control is a fundamental and difficult question for all cell types. The fission yeast* S. pombe *has a regular rod shape and divides at a reproducible cell length and a constant cell diameter. Combined with other advantages as a model system, the fission yeast is one of the best systems to study the molecular mechanisms of cell-size control. Previous studies have proposed an elegant and simple model for the regulation of cell size: Pom1 polar gradient and Cdr2 kinase-based interphase nodes work together to control mitotic entry and cell size. One line of key evidence of the model is that the Pom1 polar gradient has highest concentration at the cell pole and lowest concentration in cell middle, which decreases as cells grow. When cells reach a certain length, Pom1 gradient cannot inhibit Cdr2 kinase activity in the nodes, then Cdk1 kinase becomes active and cells enter mitosis*.

*The new work from Pan et al. proposes a new Cdr2-based model based on rigorous data analyses and mathematical modeling. They find that Pom1 concentration at cell middle does not change during cell growth, instead the Cdr2 concentration increases with cell length. They pointed out that nuclear exclusion of Pom1 accounts for the differences. The data in this high quality work are very convincing and well supported by the careful controls. I do not have major concerns for this elegant study*.

We thank this reviewer for appreciating the high quality and considerable care that has gone into this work. We point out for this reviewer that we have added significant new data on scaling with cell surface area (Figure 6) and the effects of Pom1 on Cdr2 (Figure 7). We believe these have added greatly to the significance of this work.

*The minor concerns listed below are not essential for the main conclusion of this paper*.

*1) I do not understand why the mean background outside the cells was subtracted when measuring fluorescence intensity. When measuring cortical Pom1 and Cdr2 intensity, most background in the ROI should come from diffused molecules within the cells. Thus, it might be better to use the cytoplasm as background. This will not affect the main conclusion of this study. Instead, it will make the increase of Cdr2 intensity at cell middle more obvious since Cdr2 global concentration decreases slightly when cells grow longer (*Figure 4—figure supplement 1*)*.

We have tried the analysis both ways, and use the cytoplasmic background in most of our analyses.

*2) It is impressive that six methods were used to measure Cdr2 fluorescence in cortical nodes. However, five of them are based on maximum-intensity projections. The sum-intensity projections are more suitable since the total proteins in each nodes instead of maximum pixel intensity might be most important for cell-size control. This may explain the steeper slopes of all other panel except E in*
Figure 2—figure supplement 3.

We have carefully considered these different methods of measuring fluorescence intensity. The maximum intensity measurements make sense when the nodes are considered as point sources on the cortex. The sum projections take into account the cytoplasmic and out of focus information that are less relevant here. Therefore, sum projections are considerably noisier than maximum intensity projections. The important point is that the same increasing trend in Cdr2 density is found using all the different approaches. In general, we analyzed data using both maximum and sum projections to confirm that our results were not significantly altered by the choice of method. For the reasons stated above, we typically show the results from maximum intensity projections, although we do use sum projections in measurements of whole cell intensities. We emphasize that our key conclusions are replicable with different analysis methods.

*3) In*
Figure 2*, each cell has only about 30 to 40 nodes. This might be an underestimation due to large z spacing (∼0.4 μm). To count node number accurately, much smaller slice spacing, like 0.05 or 0.1 μm, is needed*.

As suggested, we compared imaging using 0.1 µm vs. 0.4 µm Z spacing. There were no differences in the number of nodes observed. The node number might seem smaller than previous counts, because we used a thresholding approach to count the bright, mature nodes in an objective automated manner. By eye, there are many smaller nodes that are missed in this analysis, but it is difficult to objectively count their number. By eye, we can discern around 60-70, as previously reported (originally by our lab). However, we preferred the automated method to measure the increase in the number of nodes (rather than the absolute number), as the analysis is objective and easier to perform in a large number of cells.

[Editors’ note: the author responses to the re-review follow.]

*The Reviewing editor and reviewers discussed their comments before we reached this decision, and the Reviewing editor has assembled the following comments to help you prepare a revised submission. Based on these discussions, we would like you to address the two points raised below*.

*1) One of the reviewers (#2) has made a point about computing Cdr2 gradients near the SPB and it seems reasonable to attempt this given that you have the relevant information on Cdr2 concentration. The referee's point is provided verbatim*
*below:*

*“Whether increased medial Cdr2 is in fact what triggers mitotic entry is still not clear. The authors suggest that Cdr2 localization at the medial cortex could generate a concentration gradient that extends to the SPB where the Cdr2 target Wee1 resides. As Cdr2 medial concentration increases with cell elongation, it is proposed that the Cdr2 concentration at the SPB would increase until it is sufficient to inactivate Wee1. I have some concern about this idea because cytoplasmic gradients are likely to be very steep given the high rates of diffusion. Given that the authors know both the Cdr2 concentration at the medial cortex and the turnover rate they could calculate how steep the gradient from the cortex is likely to be and whether the concentration of Cdr2 around the SPB would actually change significantly as cells elongate. This would at least say whether this notion made sense. If so, it would be interesting (in future studies) to examine mutants that caused the SPB to be positioned further away from the cell cortex to see if this changed the timing of mitotic entry*.

This point is in regards to one sentence in the Discussion speculating on how the nodes might communicate to downstream targets. This downstream event is not a major focus of the paper, and our statement in the text is just a suggestion of how this might work; we do not claim that it works in this way, and other possibilities can be imagined.

Prompted by this question, we have however considered carefully if a cytoplasmic gradient is a credible means of communicating information about cell size from the nodes to the SPB.

We first note that the SPB can often be situated very close to the cortex. Images of the SPB in G2 phase frequently show that it is positioned very close to the medial cortex. Indeed, some EMs show the SPB within 50-100 nm of the plasma membrane, and so it is possible that the SPB may directly contact the cortex.

We have considered quantitatively what sort of gradient of cdr2p might be produced given the numbers obtained from our experiments. Our calculations predict a very shallow gradient, due to the high rate of cytoplasmic diffusion. Our FRAP results show t_1/2_ is ∼3 min (Figure 2—figure supplement 4), which gives an off rate of ∼4x10^-3^ s^-1^. With ∼3000 cdr2p molecules in the nodes, the total rate of cdr2p molecules released from all the nodes is ∼10 s^-1^. Assuming a reasonable diffusion constant, it would take only ∼10s for a cdr2p molecule to diffuse across the cell, before becoming homogeneously distributed in the cytoplasm. Hence, the number of molecules making up the entire cytoplasmic gradient would only be about 100. This is a very low number in comparison with the overall number of cytoplasmic cdr2p molecules, making for a very shallow gradient. Moreover, the localized SPB would only be able to sample a fraction of these gradient molecules, which would increase the noise still further. Thus, it is unlikely that this simplest mechanism based on sensing the number of freely diffusing cdr2p molecules emitted by the nodes would work.

This gradient mechanism could plausibly work, however, if some features are added. First, the gradient may be improved by phospho-regulation; in this case cdr2p emanating from the nodes may be active only for a short time, while the rest of the cytoplasmic pool is not active. Second, there could also be diffusion barriers that prevent mixing of the cdp2p from the nodes and the cytoplasmic pool. This situation may arise if the SPB and nodes were basically “touching” or surrounded by ER membranes.

For this paper, we have added to the Discussion a sentence that gradient fidelity could be improved significantly through the use of a phospho-gradient or diffusion barriers, with noise further reduced through time-averaging.

Further investigation of this process will require future experiments and modelling. We agree that it would be interesting to test the effect of microtubule-dependent SPB positioning on cell size control in the future. We have done some initial simulations of this process, but given the many unknowns here we think it best to leave a more extensive analysis for the future.

*It is not clear from the main text if the model explains the existence of medial nodes, how they are distributed, and how new ones form with*
*cell length”*

We have added 2 sentences to the paragraph ‘Mathematical models for size-dependent accumulation of Cdr2p nodes’ in the main paper to make it clear that the model does not specifically incorporate the details of nodal formation/growth. Rather we model the total number of cdr2p molecules in the nodal region. Currently we lack sufficient information on the detailed dynamics of individual nodes in order to more precisely model their formation and growth.

*2) Please also address the following point from*
*referee #3:*

*“One minor improvement required for new figures (and some old figures) is that most Y-axes in the graphs should*
*start from zero.”*

We have re-plotted many of the graphs in Figure 7, and corresponding supplement, to amend the Y-axes, as requested. However, we have left Figure 6 unaltered, as the significant differences in these plots would become less clear on plots with Y-axes starting from zero.